# HOH: Markerless Multimodal Human-Object-Human Handover Dataset with Large Object Count

**Noah Wiederhold**
Clarkson University
wiedern@clarkson.edu

**Ava Megyeri**
Clarkson University
megyeram@clarkson.edu

**DiMaggio Paris**
Clarkson University
parisda@clarkson.edu

**Sean Banerjee**
Clarkson University
sbanerje@clarkson.edu

**Natasha Kholgade Banerjee**
Clarkson University
nbanerje@clarkson.edu

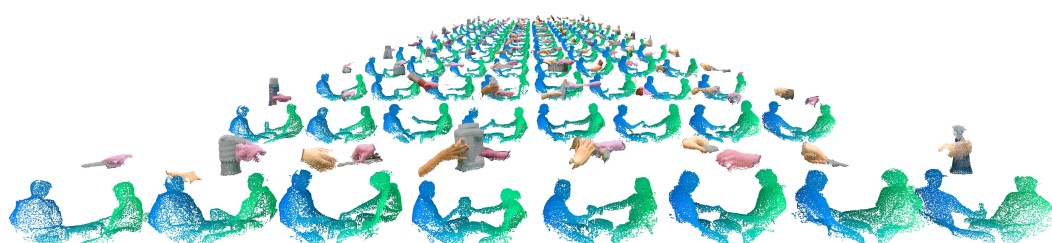

Figure 1: HOH is a markerless 3D multimodal dataset on human-human handovers with 136 objects and 20 participant pairs, 40 accounting for role-reversal. We show 3D point clouds of the upper body, giver (pink) and receiver (yellow) hands, and objects (colorized) fused from 4 Kinects from various time points of handover and across various participant pairs and objects. The dataset demonstrates diversity of object geometry, participant posture, interaction approaches, and grasp types. Shown are right-seated givers. The dataset also consists of left-seated givers.

## Abstract

We present the **HOH** (**H**uman-**O**bject-**H**uman) Handover Dataset, a large object count dataset with 136 objects, to accelerate data-driven research on handover studies, human-robot handover implementation, and artificial intelligence (AI) on handover parameter estimation from 2D and 3D data of two-person interactions. HOH contains multi-view RGB and depth data, skeletons, fused point clouds, grasp type and handedness labels, object, giver hand, and receiver hand 2D and 3D segmentations, giver and receiver comfort ratings, and paired object metadata and aligned 3D models for 2,720 handover interactions spanning 136 objects and 20 giver-receiver pairs—40 with role-reversal—organized from 40 participants. We also show experimental results of neural networks trained using HOH to perform grasp, orientation, and trajectory prediction. As the only fully markerless handover capture dataset, HOH represents natural human-human handover interactions, overcoming challenges with markered datasets that require specific suiting for body tracking, and lack high-resolution hand tracking. To date, HOH is the largest handover dataset in terms of object count, participant count, pairs with role reversal accounted for, and total interactions captured.

37th Conference on Neural Information Processing Systems (NeurIPS 2023) Track on Datasets and Benchmarks.

# 1   Introduction

Human-human handover of objects is a complex process that has been highly studied due to its role in enabling fluent human-robot interaction (HRI) in collaborative operations. Researchers have investigated a wide range of parameters underlying handover, including physical factors such as grip force [40, 16, 20, 19, 31], interpersonal distance [4, 27], object orientation [14, 15], object mass [27], hand movements [48, 46], and reaction times [23], as well as cognitive parameters such as giver/receiver intent communication [38], gaze variation and joint attention for shared goals [50, 42, 55], movement adaptation [28], and affordance preferences [18]. The study of these parameters has given rise to a plethora of work in data-driven human-robot handover interactions [44, 36, 30], and has shown potential for learning-based prediction of handover parameters such as receiver grasp [54]. Given the often non-verbal aspects of handover and its dependence on interpersonal coordination for shared goal success [33], there is a strong interest within the cognitive science community [33] to understand "What Does a Handover Tell" [6], e.g., sensitivity of motion kinematics to social intention [5], potential individuality of short-range handover trajectories [6], and inter-person coordination. The study of handover also has a societal benefit, of providing robots that engage in patient-centric caregiving and productivity-aware collaboration.

The large-scale propagation of human-human handover research has been hindered by two challenges. The first challenge is that, so far, human-human handover studies have used **small object counts**, typically 2-5, a few with 10-20 [15, 13, 35], and to-date no more than 30 [54]. Given the vast diversity of objects likely to be interacted with by robots in consumer spaces—tools, kitchen utensils, containers, toys, fruit, bathroom items, office supplies, and electronic items to name a few, conclusions on physical and cognitive parameters in small object count studies cannot be generalized to objects in the wild, without fully studying the impact of variation in object properties such as size, shape, mass, functionality, and presence of protrusions or affordances.

The second challenge is that publicly available datasets [15, 13, 31, 35], apart from being few and using small object counts, use **marker-based motion capture (mocap)** with only a single marker on the wrist and 1-5 markers on the object. Markered body mocap suffers from known limitations such as use of form-fitting suits that prevents clothing diversity, and lack of high-resolution hand geometry and object structure that prevents analysis of spatial affordance during grasp and object transfer [18] to enable safe human-robot handover. Attempts have been made to use public datasets to develop human-inspired robotic controllers [36, 21, 30], demonstrating their value. However, the constrained markered setup and small object count hinder their use in studying parameters of natural handover or use in developing learning algorithms for robotics that rely on large data with high degree of diversity.

We contribute **HOH** (**H**uman-**O**bject-**H**uman), the first markerless, high object count human-human handover dataset that is publicly available. HOH contains 2,720 interactions performed by 40 participants organized in 20 role-reversing giver-receiver pairs covering a total of 136 objects—116 store-bought and 20 3D printed—spanning 17 form/function categories and 8 everyday use classes. We adopt a markerless approach to capture natural real-world motions and clothing. We use a multi-camera setup of 4 30FPS Kinect RGB-D sensors and 4 60FPS FLIR Point Grey cameras to perform 360° allocentric (non-body-mounted) capture of human-human handover. We record post-handover giver and receiver perceptions of comfort. We provide the following contributions in HOH.

1. Kinect depth video for 2,720 handovers from 20 participant pairs, and Kinect and Point Grey color video for 2,448 handovers from 18 pairs who consented to identifiable color (IC) data release.
2. Manual ground truth (GT) annotations of key events on first giver grasp on object, object transfer, and last receiver contact, giver hand, object, and receiver hand masks assisted by Segment Anything Model (SAM) [32], and giver and receiver handedness and grasp type at object transfer using the taxonomy of Cini et al. [18].
3. Processed data in the form of full 360° point clouds (color mapped for IC pairs), OpenPose [12] skeletons, tracked hand and object masks, and color-mapped hand and object point clouds over from first to last key event frame in all 2,720 interactions.
4. Information on the object used in each interaction, including mass, class, category, and 3D model.
5. Object 3D model GT alignments to frames ranging from the first key event to the last key event, providing GT 6DOF object pose.
6. Analysis of object, trajectory, and grasp properties.
7. Experimental results of neural networks trained to predict giver grasp and transfer orientation using object point clouds, and receiver grasp and trajectories using object and giver data.

**Intended Use Cases.** The dataset facilitates studies on human-human handover interaction with a broad range of objects to inform cognitive psychology and human-robot handover research. The dataset enables investigation of parameters such as giver and receiver kinematics, object motions, giver/receiver hand and upper body coordination for shared goal accomplishment, grasp type, multi-person handedness, and key event timing. Investigations can include per-parameter statistical analyses, cross-parameter relationships, and relationships with respect to factors such as object geometry, participant demographics, and subjective ratings of comfort. The benefit of 3D markerless capture for HRI is to develop learning-based algorithms that inform robotic manipulators connected with RGB-D sensors on estimating where to grasp to enable safe human-robot handover using point clouds backprojected from RGD-D data. Robot givers can learn from the behavior of human givers where to preferentially hold objects, how to move during giving, and how to orient the object at the transfer point when handing the object to a receiver. Robot receivers can use the behavior of human receivers to learn where to grasp objects handed to them by a giver, and what trajectory to navigate in space to remain safe. By containing full upper-body capture and associated skeleton estimation, the dataset enables the study of relationships such as hand and body articulation, and head pose and eye gaze at various key points of the handover. These analyses are motivated by prior human-robot handover implementations [28, 36, 30, 42, 55, 48, 46] and cognitive studies [5, 6] based upon similar human-human handover studies on smaller object sets. Study of simultaneous giver and receiver comfort perceptions is expected to enable understanding of alignment between the giver and receiver, facilitating study of inter-person coordination important to shared goal accomplishment [34].

## 2 Related Work

**Human-Human Handover Datasets.** To the best of our knowledge, 4 publicly available human-human handover datasets exist at this time [13, 15, 31, 35]. As discussed in Section 1, they are hindered by the limited object quantity, and low-resolution data and unnatural interaction setting due to use of markered mocap. Khanna et al. [31] focus on grip force measurement, and use a single unweighted and weighted force-sensor instrumented baton to gather mocap data. Their data prevents understanding handover dependence on object geometry. Carfi et al. [13] and Kshirsagar et al. [35] provide RGB-D data from 1 and 2 views respectively. The Kinect FOV makes fewer than 4 views insufficient for full 360° coverage, e.g., to capture hand-object surfaces occluded from the camera view. Though Chan et al. [15] provide raw color data from their Vicon cameras, they use 8 views, lack depth data, use a wider-out setup, and have participants wear black body suits. The low view count, lack of texture, and high distance is unlikely to yield success in using multi-view stereo for 3D reconstruction. Their color data is in a proprietary Vicon format requiring a license purchase.

We have identified two more multi-object dataset collections [18, 54] that are not publicly available. Cini et al. [18] perform grasp taxonomy analysis by recording mocap using IR reflectors on the hand dorsum and object, with one video per interaction. Their data suffers from similar concerns as the public datasets. Ye et al. [54] acquire mocap using 6 magnetic markers per hand (5 at fingertips and one at wrist) and 3 optical markers per object, as well as RGB-D video using 5 30FPS allocentric cameras. They provide deep networks to hand-object pose estimation and giver-conditioned receiver pose prediction. The 5-fingertip capture provides higher detail than other datasets, however, knuckle articulation is absent. Though the RGB-D cameras can provide dense detail, the magnetic sensor cables occlude the hand dorsum. The RGB-D views lack face and upper body data, important to analyze gaze [42, 21] and arm extension. To ensure comprehensive natural grasp and structure capture, we use a markerless allocentric setup with 360° coverage of the upper body of both participants.

We summarize key properties of multi-object datasets in comparison to HOH in Table 1. HOH is the largest multi-object dataset in terms of object counts, participants recruited, pairs with role-reversal (RR) accounted for (matching the non-public H2O dataset), and number of cameras (matching the count in Chan et al. [15]). Our object count at 136 is 6.8× the count of Chan et al., the public dataset with the next largest count. 10 participant pairs (20 with RR) in HOH interact with 68 objects and the other 10 pairs (20 with RR) interact with a separate set of 68 objects, resulting in multi-pair interactions for the same object set with 3.4× the object count in Chan et al. HOH is the only dataset with simultaneous giver and receiver comfort ratings.

**Hand-Object Datasets.** In consisting of grasp-based hand-object interactions, human-human handover shares features with the large body of recent work in hand-object interaction, focused

Table 1: Comparison of HOH (last column) versus prior multi-object human-human handover datasets. RR = role reversal, i.e., giver becomes receiver and vice versa, and is only applicable for some datasets. SB = Single-blind (⋆experimenter is one participant), DB = Double-blind (both participants are recruited). †from Vicon X2D. **Obtained by placing markers on object, rather than by model alignment.

| | Cini [18] | Chan [15] | Carfi [13] | | Kshirsagar [35] | | H2O [54] | HOH |
|---|---|---|---|---|---|---|---|---|
| | | | SB | DB | Biman. | Uniman. | | (Ours) |
| # Interactions | 1,734 | 1,200 | 799 | 288 | 240 | 120 | 1,200 | **2,720** |
| # Objects | 17 | 20 | 3 | 7 | 10 | 5 | 30 | **136** |
| # Participants | 34 | 20 | 18 | 18 | 24 | 24 | 15 | **40** |
| # Pairs | 17 | 10 | 18⋆ | 9 | 12 | 12 | **40** | 20 |
| # Pairs with RR | 17 | 10 | 36⋆ | 18 | 24 | 24 | **40** | **40** |
| # Cameras | 1 | **8** | 1 | 1 | 2 | 2 | 5 | **8** |
| Markerless? | ✗ | ✗ | ✗ | ✗ | ✗ | ✗ | ✗ | ✓ |
| Ratings? | ✗ | ✗ | ✓ | ✗ | ✗ | ✗ | ✗ | ✓ |
| Color? | ✓ | ✓† | ✓ | ✓ | ✓ | ✓ | ✓ | ✓ |
| Depth? | ✗ | ✗ | ✗ | ✗ | ✓ | ✓ | ✓ | ✓ |
| Point Clouds? | ✗ | ✗ | ✗ | ✗ | ✗ | ✗ | ✗ | ✓ |
| 3D Object Model? | ✗ | ✗ | ✗ | ✗ | ✗ | ✗ | ✓ | ✓ |
| 6DOF Object Pose? | ✗ | ✓** | ✗ | ✗ | ✓** | ✓** | ✓** | ✓ |
| Public? | ✓ | ✓ | ✓ | ✓ | ✓ | ✓ | ✗ | ✓ |

on, e.g., AI-driven hand and object pose estimation in the presence of hand-object occlusions, and even locations for safe robotic grasp in human-to-robot handover [17]. Much recent work has been invested in the collection of **single-person hand-object interaction datasets** [56, 9, 51, 17, 8, 29, 39, 25, 37, 26, 49, 22]. AI algorithms and research findings from single-person hand-object interaction datasets, even when bimanual, cannot be directly transferred to handover interactions involving *two different people*. Inter-participant differences introduce variations in hand geometry and appearance. Spatial kinematics vary across the two types of interactions. Handovers may cover larger translations during reach and transfer, making egocentric setups [37, 39] infeasible due to the constrained view field. Single-person bimanual interactions involve mirrored right and left hands. The preponderance of right-handed individuals means that handovers with unimanual interactions are likely to be dominated by non-mirrored right-right hand interactions at the giver-to-receiver object transfer phase. These concerns justify the need for a distinct handover dataset such as HOH, in which over 75% of interactions occur using non-mirrored hands. Several interactions in HOH have 3+ hands, e.g., unimanual giver grasp to bimanual receiver grasp. HOH exceeds all but the egocentric-only HOI4D dataset [39] in object count. Though Assembly101, with 101 objects, approaches HOH, Assembly101 lacks paired and aligned 3D scans.

## 3 HOH Dataset

### 3.1 Dataset Design Philosophy

Motivated by the goal of 360° capture of natural handover, we used a markerless allocentric capture approach and recorded giver-to-receiver handover in a seated pose to minimize fatigue. Seated handover interactions are common in social settings, e.g., at a coffee shop or in a collaborative work environments.We employ the standardized nomenclature of Kopnarski et al. [33] who define handover as consisting of four phases, *reach and grasp*, where the giver applies enough grip force for object hold, *transport* where the object is moved to reach the shared transfer point, *object transfer* during which the object is handed over from giver to receiver, and *end of handover*, during which the receiver acquires full object possession and "uses the object in line with their intention" [33]. In our study, end of handover involves placing the object on the table.

Some prior studies inform participants to consider intended use, or specifically request bimanual or unimanual grasp [35], or giver/receiver consideration [15]. We provide no prompting regarding interaction, except that the giver maintain grasp during transport. Our goal of unprompted interaction was to acquire natural giver reach and grasp, similar to the *natural* condition of Chan et al. [15], to enable analysis of grasp type and hand combinations in open-ended handover, and to avoid

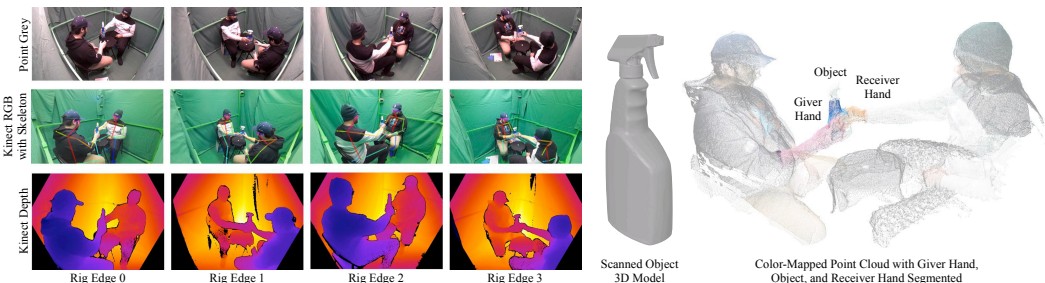

Figure 2: Left: Synchronized color and depth data collected in HOH from 4 viewpoints for a handover interaction, with OpenPose skeletons show in in Kinect color images. Center: Object 3D model used in interaction. Right: Color-mapped 3D point cloud with segmented giver, object, and receiver.

Table 2: HOH Dataset Summary. GT = Ground Truth Annotation, Pre = pre-GT processing, post = post-GT processing to propagate annotations. *Currently all skeleton estimation, color-mapping, segmentation, GT, and post-processing has been conducted for Kinect images.

| | Modality | Modality Description | Interactions | Count |
|---|---|---|---|---|
| **Pre** | Kinect Color | 4-viewpoint 30FPS RGB Video, 1920x1080 | 2,448 | 1.4M |
| | Kinect Depth | 4-viewpoint 30FPS Depth Video | 2,720 | 1.6M |
| | FLIR Point Grey Color | 4-viewpoint 60FPS RGB Video | 2,448 | 2.8M |
| | Skeletons* | Estimated using OpenPose [12] in Kinect color images | 2,720 | 1.6M |
| | Fused Point Clouds | Fused from depth images using multi-camera calibration | 2,720 | 250K |
| | Fused Color Point Clouds* | Colorized using Kinect color images | 2,448 | 240K |
| | Full Color Segmentation Masks* | Extracted using SAM [32] | 2,720 | |
| **GT*** | Key Event Annotations | Frames for first giver-object contact (G), giver/receiver grasp at transfer (T), and last receiver-object contact (R) | 2,720 | 8,160 |
| | Giver Hand / Receiver Hand / Object Segmentation Masks | Manually isolated from SAM segmentation for giver hand from G and T, object from all G, T, R, and receiver hand from T and R, done for as many viewpoints where entity is visible | 2,720 | 8,513 / 8,879 / 21,224 |
| **Post*** | Tracked Giver Hand / Receiver Hand / Object Masks | Tracked using Track Anything from G to R | 2,720 2,720 | 710K / 780K 1M |
| | Giver Hand / Receiver Hand / Object Point Clouds | Segmented from fused point clouds using tracked masks | 2,720 2,720 | 240K / 260K 290K |
| | 3D Model & 6DOF Object Pose | Aligned using iterative closest point (ICP) [7] from 3D model to G, G to G+1, G+1 to G+2, · · · R-2 to R-1, and R-1 to R. | 2,720 | 290K |

compromising data diversity, a concern for learning-based algorithms. Given the unstructured nature of the interaction, a giver could be inconsiderate of the receiver's comfort, or in thinking of the receiver could compromise their own comfort. We obtained a 7-point Likert scale rating after each interaction to gauge giver and receiver comfort.

Our data is captured in a 1.7m × 1.7m × 2.0m green-screened T-slot frame rig using 4 Azure Kinect RGB-D sensors and 4 FLIR Point Grey BlackFly S high-speed color cameras, as shown by the images in Figure 2. The supplementary covers the setup, computing, connection, and networking setup and details to conduct simultaneous multi-sensor capture, spatial geometric camera calibration, and post-capture synchronization via an overhead light. In 160 of the 2,720 interactions (4 per pair), we requested the participants to wear blue gloves with the intention of acquiring high-fidelity tracking using color-segmentation. The Segment Anything Model (SAM) [32] now facilitates tasks such as background-region removal, and foreground extraction as performed for this dataset. However, we collected HOH data prior to SAM, when reliable off-the-shelf easy-to-use segmentation tools were unavailable. We opted for a close-confined setup over spread-out to acquire color and depth data at the highest resolution feasible using Kinects and Point Grey cameras.

## 3.2 Dataset Collection

**Dataset-in-a-Nutshell.** HOH contains multimodal data on handover interactions from a total of 20 participant pairs, or 40 pairs with giver-receiver role-reversal (RR), interacting with a total of 136 objects while seated in our multi-camera capture space. 10 pairs interact with 68 and the other 10 pairs with the remaining 68. Table 2 provides a summary of HOH data. We pair HOH interactions with a 136-object dataset containing 3D models and metadata.

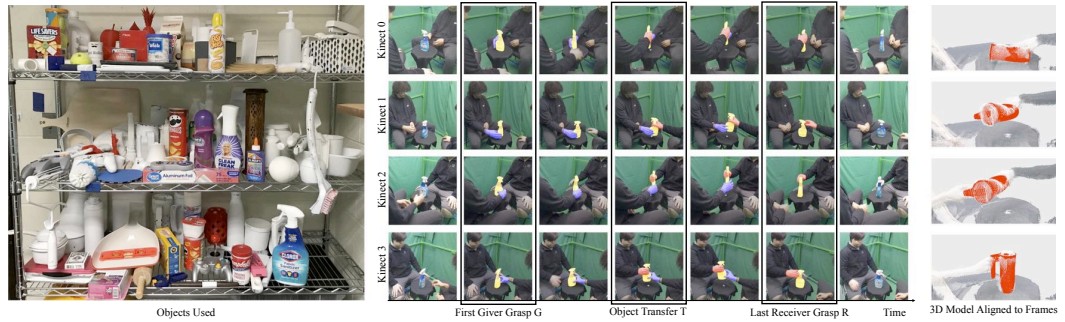

Figure 3: Left: Objects used in HOH. Center: GT annotation of key events, and SAM and GT-aided tracks of giver hand (purple), object (yellow), and receiver hand (red) segments. Right: 6DOF pose estimation via ICP-alignment of 3D model (red) to frames.

**Object Dataset.** HOH's diverse 136-object dataset, shown in Figure 3, spans 8 everyday use classes—toys (14), mugs (11), food and drink items (13), cooking utensils (25), tools (15), office items (11), household items (27), and 3D printed items (20). Objects are organized into 17 categories based on aspect ratio and functionality, with 8 objects per category. 15 categories span 3 aspect ratio (or form) bins—1:1-2:1, 2:1-3:1, and >3:1—and 5 bins that describe whether the object has (H) or lacks (NH) a handle, has (F) or lacks (NF) an end used for a function such as cutting, drinking, peeling, or screwing, and is found in a vertical (V) or horizontal (Z) standard orientation. The 5 bins are NFNHZ, NFNHV, FNHZ, FNHV, and FHV. Category 16 consists of 1:1 aspect ratio objects in FHZ, and category 17 consists of objects outside categories 1 to 16. We found that FHZ objects with a larger aspect ratio are uncommon. Each pair of participants interacted with 4 objects per category to cover their allocated 68 objects while ensuring full category spanning. The supplementary provides details on object preparation and acquisition.

**Participant Recruitment.** We recruited 40 participants from the local university population, through message posting on online collaborative work spaces, after receiving approval from the university's Institutional Review Board (IRB). Participants were not compensated for providing data. The research did not involve contract work or crowd-sourcing. We had 34 male and 6 female participants, organized into 6 female/male pairs, and 14 male/male pairs. No two pairs had the same participant. 2 participants declined to share their age. Ages of remaining participants ranged from 19 to 51, with mean of 24.8±7.4. Height ranged from 1.55m to 1.96m with mean of 1.8m±0.1m. 38 of our participants reported writing right-handed, only 1 reported being a left-handed writer, and 1 reported being ambidextrous. 2 participants have not given consent for sharing identifiable color (IC) information. For safety, at this time, we exclude all color frames for all their interactions in public release, i.e., color data from 2 pairs, even if some color frames do not identify their face.

**Experiment Procedure.** Participants were informed through the Informed Consent form that the experiment was minimal risk, participation was voluntary, participants could wear masks for COVID-19 safety and must consent to mask wearing if their partner wished it, no identifiers would be collected, and participants could request that IC images be publicly inaccessible. Participants filled a demographics questionnaire, were assigned a random 5-digit ID, and were introduced to their partner. Participants were assigned giver and receiver roles. Upon sitting at the setup, the participants performed handovers with 68 objects randomly ordered. Participants wore gloves for a random set of 4 handovers. The experimenter placed the object at tabletop center in a random orientation. Apart from committing to grasp and receiver placing object on the table, participants were free to perform interactions as they wished. Both participants filled out post-handover 7-point Likert rating on comfort using a clipboard. Participants then reversed roles, and re-performed handovers with the same 68 objects in a different order. In all but two cases, participants returned to the same seats, resulting in 22 and 18 RR pairs with left- and right-seated givers respectively.

**Data Pre-Processing.** We processed recorded camera data to obtain time-synchronized color and depth images discussed in Table 2 over all views, ensuring that dropped frames are accounted for if any. We backprojected the depth pixels into 3D using the depth image intensity values. We use the transformations between the depth cameras to the reference camera 0 to fuse backprojected pixels into a single point cloud per frame number oriented in camera 0. We used depth-to-color transforms

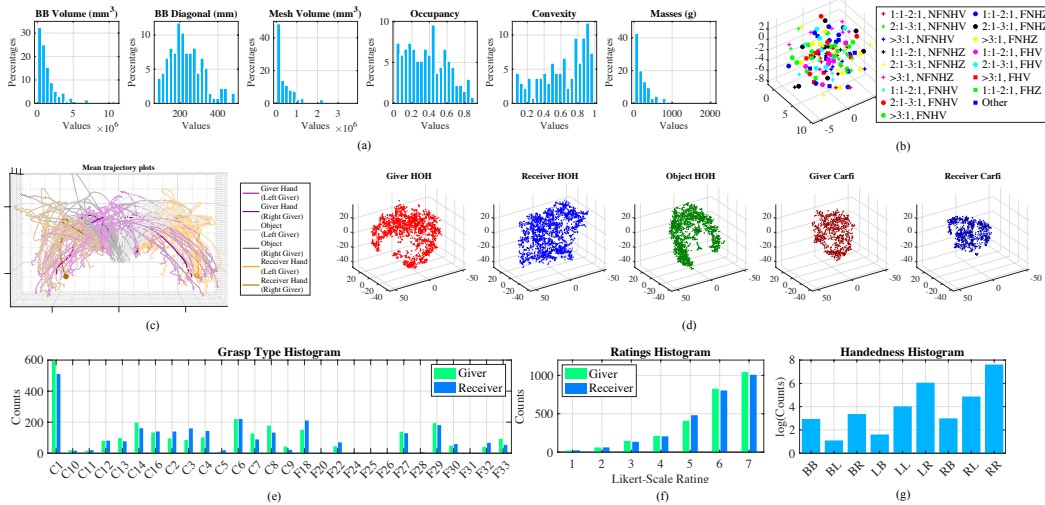

Figure 4: Dataset Analysis: (a) histograms of object properties, (b) t-SNE for object models, (c) Overall (dark) and per-pair (light) mean trajectory plots with start points marked using spheres, (d) t-SNE of trajectories for HOH compared to Carfi et al. [13], (e) histogram of grasp types per Cini et al. [18],(f) histogram of grasp ratings, and (g) log histogram of handedness counts at transfer.

to color-map the fused point clouds. We ran OpenPose [12] and SAM over all Kinect color images to acquire upper body skeletons and complete image segmentation masks.

**GT Annotation and Verification.** As shown in Figure 3, we manually annotated three frames—G, T, and R—marking key events. G contains first giver contact marking the grab portion of reach and grab, T contains simultaneous giver and receiver grasp marking the middle region of object transfer, and R contains last receiver contact on the object marking the end of handover. We marked SAM masks for three entities —giver hand, object, and receiver hand—in G and T for giver, G, T, and R for object, and T and R for receiver. Masks were marked when present and multiple masks per entity are fused into a single mask. We verified and corrected all SAM masks. We checked all OpenPose skeletons and found that only 4.6% had either missing or inaccurate joints. We store OpenPose confidence values. We labeled handedness and grasp type using the 28-class grasp taxonomy of Cini et al. [18].

**Post-Processing.** We used Track Anything [53] to track annotator-segmented SAM masks from frames G to R. We backprojected tracked masks via the camera calibration to generate giver, object, and receiver point clouds. We denoised full, giver, object, and receiver point clouds to remove outliers, and use median filtering based on the top $k$ nearest neighbors per point to smooth the colorization, with $k$ being 10. We save both pre- and post-cleaned point clouds.

**6DOF Object Pose.** We provide GT object 6DOF pose by aligning the object 3D model to all frames from G to R for all handover interactions. First, we conduct a frame-to-frame ICP [7] alignment of the object point cloud to estimate inter-frame cloud-to-cloud transformations of the object point cloud for all frames, linking all frames from G to R. Next, we conduct an automated alignment of the 3D model to the object point cloud in the G frame. We sample a diverse range of orientations to ensure exhaustive $SO(3)$ coverage. We transform each 3D model using each orientation and fine-tune the transformed model's alignment to the object point cloud using ICP. We choose the 3D model alignment with the smallest ICP distance. We use the inter-frame transformations to align the 3D model to all frames from G to R. Example frames from the alignment are shown in Figure 3.

### 3.3 Dataset Analysis

**Objects.** Figure 4 provides a summary of multi-modal analysis of our dataset. Object histograms in Figure 4(a) show that while our objects are skewed toward lower bounding box volume (BBV), mesh volume (MV), and mass (medians of $9.73 \times 10^6 \text{mm}^3$, $2.54 \times 10^6 \text{mm}^3$, and 167g, and skewness of 2.83, 2.77, and 3.47 respectively), the bounding box diagonal is normally distributed with mean of 233.58mm±88.50mm, and 17 objects are over 0.5kg. The bounding box occupancy (MV-to-BBV

ratio) is slightly skewed toward lower values (median 0.37, skewness 0.27) indicating that HOH has high count of non-full objects, e.g., the spray bottle in Figure 2. The histogram of convexity, computed as MV-to-convex-hull ratio [2], shows though our dataset tends toward convex objects (median 0.67, skewness -0.51), 42 objects have a convexity < 0.5, i.e., embody concavities. The spread of coefficients t-SNE plot of object geometry, in Figure 4(b), computed using 8,000 3D point samples per 3D object model surface, confirms the object diversity in our dataset.

**Trajectories.** Figure 4(c) shows overall (dark) and per-pair (light) mean trajectories for giver hand, object, and receiver hand. Means are obtained by resampling trajectories to have 100 samples, and averaging within left-seated and right-seated givers. Trajectory means show transport and object transfer phases for giver hand, inclination of object toward receiver indicative of giver intention to participate in collaborative handover, giver retraction after transfer, and looped arc for receiver toward end of handover. Left- and right-seated givers mirror each other. Spread of per-pair means demonstrates trajectory diversity, echoing prior findings of individual dependence in handover trajectory [6]. HOH trajectory diversity is confirmed by the t-SNE plot in Figure 4(d) where the coefficients show a higher spread than the trajectories of Carfi et al. [13]. To ensure comparability across HOH and Carfi trajectories, we align the trajectories within each entity and dataset using Procrustes prior to t-SNE computation, since participants may be spatially displaced in the Carfi dataset. We find trajectory lengths follow a normal distribution with giver, object, and receiver means of 3.26s±0.97s, 3.60s±0.94s, and 3.49s±0.95s. Object trajectories are longer since the receiver may react slightly later than the giver, while the giver may complete their motion before the receiver's end of handover.

**Grasp.** The distribution of grasp types in Figure 4(e) organized by the taxonomy in Cini et al. [18] shows a majority of power grasps especially near C1, or larger diameter grasps apt for HOH object sizes. High precision grasp counts are found for the giver, e.g., C6/C8 thumb-finger and C14 tripod grasp, common for rod-like or handled-equipped items. For the receiver, we see high counts for C6, F18 extension-type grasp on thin flat items, and F29 or stick grasp likely during receipt of objects with protrusion affordances. The supplementary details taxonomy nomenclature. The comfort rating histogram in Figure 4(f) indicates skew toward higher comfort levels (giver comfort skewness of -1.23 with 16.21% over 4, receiver comfort skewness of -1.21 with 15.7% over 4). We observe a high overall giver-receiver Pearson correlation of 0.38. As shown by the log histogram for handedness in Figure 4(g), for unimanual grasps the giver and receiver use the same hand in 76.8% of the interactions, and opposite hands in 20.4%. 76 interactions have bimanual grasp.

## 4    Experimental Results Showing Use Case

The primary use of our dataset is to drive AI research in HRI. Unfortunately, no off-the-shelf algorithms exist to directly evaluate multi-person data such as HOH for robotic understanding of handover parameters. The only approach for learning-driven grasp lacks public code and data [54] and requires access to articulated hand pose, making it unusable with HOH. Full-fledged HOH-data-driven robotic control is outside the scope of this paper (and in fact consists of components spanning a wide range of future work). In this section, we create and evaluate **deep neural networks** to show the benefit of HOH data for **four tasks of relevance to the robotic manipulation pipeline:** (1) Use object point cloud to predict human giver grasp point cloud or o2gg, (2) Use object point cloud to predict object orientation at transfer point or o2or, (3) Use object and giver point cloud to predict human receiver grasp point cloud or g2rg, and (4) use human giver hand motion to predict receiver motion trajectory or g2rt. o2gg and g2rg enable robotic givers to bias grasp near preferred human giver grasp, and robot receivers to bias grasp away from / close to preferred human giver / receiver grasp. o2or enables robot manipulators to present objects at transfer in human-preferred poses. g2rt enables robot receiver motion planning in response to human giver motion.

**Implementation.** We adapt PoinTr for o2gg and g2rg and Informer for g2rt. We use PointNet [47] with 2 dense layers to generate quaternion orientation for o2or. We obtain input point clouds for o2gg and o2or from Gpre, and for g2rg from Tpre, where Gpre and Tpre are a few frames prior to G and T, containing object only and object+giver hand respectively. Outputs for o2gg, g2rg, and *o2or* consist of giver hand point cloud at G, receiver hand point cloud at T, and rotation from 6DOF pose at T represented as a quaternion. We train two networks each for o2gg, o2or, and g2rg, one that uses **complete** input point clouds enabling handover parameter analysis with 360° access to geometry through, e.g., multi-view fusion, and one that uses **partial** input point clouds, emulating

Table 3: Metrics for experimental results.

| Task | Metric | Complete | | Partial | |
|------|--------|----------|---|---------|---|
| | | GT | Best to Object | GT | Best to Object |
| o2gg | CD | 0.483±0.187 | 0.135±0.078 | 0.135±0.052 | 0.277±0.188 |
| o2or | MEAE | 0.851±0.253 | 0.365±0.153 | 0.843±0.227 | 0.411±0.136 |
| g2rg | CD | 0.539±0.648 | 0.147±0.182 | 0.206±0.121 | 0.288±0.275 |
| g2rt | MAE | 0.160±0.070 | 0.107±0.037 | N/A | N/A |

| Task | Metric | Complete | Partial |
|------|--------|----------|---------|
| o2gg | %OLGO | 17.39%±10.37% | 58.12%±19.46% |
| g2rg | %OLRG | 1.44%±4.73% | 1.13%±3.87% |

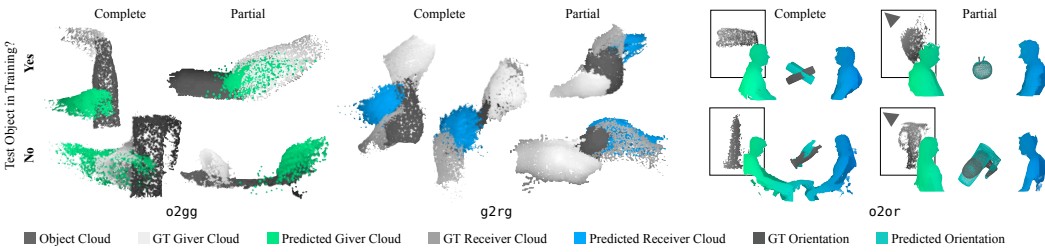

Figure 5: Visualization of outputs from o2gg, g2rg, and o2or. Insets for o2or show input point cloud. Camera angle shown for partial clouds.

single-viewpoint RGB-D sensors. We generate partial data by rendering each scene from 6 randomly generated viewpoints per scene. We pre-register all GT input and output point clouds for the complete networks to the Gpre frame, in order to assess pre-movement parameter estimation. Partial point clouds are left in the rendered viewpoint and object orientation at the appropriate handover timepoint to emulate real time behavior. Since the proposed g2rt focuses on point trajectories rather than objects, we use the trajectory centroids from Section 3.3 to train a single version of g2rt. We manually pre-transform GT to set the origin at the table center, $xz$-plane as ground, and right-seated givers aligned with left-seated.

**Training.** We randomly divide the 8 objects in each form/function bin into sets A, B, and C with 3, 3, and 2 objects per bin or 51, 51, and 34 total objects to be used in train only, test only, and train+test respectively. The training set uses random 75% of data from sets A and C. The test set uses the remaining 25% of data from set C, and 100% from set B. For g2rt, given the potential for time-varying person movement to be a behavior signature, we keep train and test participant pairs mutually exclusive. We use data from 11 pairs (22 with RR) in the training set, and the remaining 9 (18 with RR) in the test set. Hyperparameter choices and computing details are included in the supplementary.

**Evaluation Metrics.** We provide GT evaluation metrics, particularly Chamfer Distance (CD) for o2gg and g2rg, mean Euler angle error (MEAE) for o2or, and mean absolute error (MAE) for g2rt. As plausible parameters may not correspond to GT, we report best GT metrics to all test object instances (accounting for symmetry), to increase the chance of finding similar parameters. We report percentage overlap of giver hand with object (%OLGO) for o2gg and receiver hand with giver hand (%OLRG) for g2rg to gauge affordances for robot givers and receivers for safe handover.

**Results.** Table 3 shows the results of the evaluation metrics. Figures 5 shows qualitative results of predictions for o2gg, g2rg, o2or, and g2rt, the latter two contextualized within the handover providing the source input. Additional results are provided in the supplementary. When predictions diverge from GT, they correspond to plausible outputs, e.g., though the actual giver grasp for the grill brush at the bottom left of Figure 5 for o2gg is on the handle, the predicted grasp is on the brush. Variability receiver hand orientation is observed for g2rg. For o2or, we notice that the alignment of the predicted object when deviating substantially from GT, corresponds to a plausible extension direction during transfer. Outputs of g2rt show that, even without object structure, simple trajectory prediction enables the receiver to meet the giver trajectory near the shared space. From Table 3, we

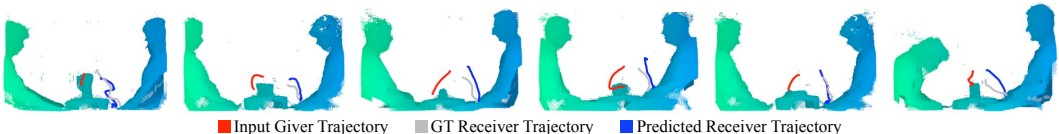

Figure 6: Visualization of outputs from `g2rt` for various pairs and objects.

notice that the mean best-to-object metrics show a drop compared to the GT, indicating that a more likely candidate for each parameter is found in the dataset, and demonstrating the dataset's diversity in representation of handover. %OLGO in Table 3, indicates affordance availability, especially using complete data. %OLRG demonstrates that giver-receiver overlap is negligible, showing ability of `g2rg` to predict away from giver hand and enabling its use for robot receiver grasp biasing.

## 5   Discussion

**Future Work.** The experimental results presented in Section 4 provide a starting point for algorithm development in HRI using HOH. The diverse, multimodal, and richly annotated data in HOH greatly opens the scope for large-scale AI algorithms in HRI. Algorithms can be developed to leverage comfort ratings to rank grasp, pose, and path prediction outputs, conduct realtime object-dependent robotic giver motion prediction and proactive generation of robotic receipt motion by giver motion forecasting, and enhance the robotics pipeline with object segmentation and tracking algorithms in the presence of multi-person occlusions due to handover. Transfer position estimation can be conducted by using upper body point clouds and/or skeletons to establish spatial relationships. The analysis in Section 3.3 provides an overall view of HOH. A large scope exists for future studies involving detailed analyses of alignment between giver and receiver comfort, trajectory velocity and timing, and coordination, with respect to grasp type, object categories, and participant pairs.

**Limitations.** HOH lacks grip force due to the use of instrumentation-free setup. Currently, grip force is collected using heavily-designed batons [40, 16, 19, 30] that constrain grasp type, are difficult to control for weight, and lack geometric diversity. The scope remains to instrument everyday objects with reliable minimally-invasive grip force units. Our participants occupy a narrow range over age and ability. We recognize the challenges in recruiting participants such as children, older adults, and individuals with different abilities, health and social concerns toward mutual interactions with unfamiliar partners, and potential cognitive barriers toward providing informed consent. Future collections can benefit from best practices on population-specific recruitment [41, 11, 3]. Though HOH has 6DOF object pose, it currently lacks GT hand pose annotation. Full manual annotation of hand pose in markerless data in the presence of occlusions and motion is a daunting task. Future work includes adapting HOnnotate [25] to operate with multi-person hand interactions.

**Societal Impacts.** HOH provides the **societal benefit** of informing social robots on how to perform safe handover, important toward the development and enhancement of trust and collaborative goal accomplishment [24, 45]. Social robots aware of object-dependent handover improve fluidity of post-handover operations involving multi-object use such as assembly and activities of daily living. Improved trustworthy social robots, engaging in safe handover of objects have the potential to fill current shortages of in-home aides for older adults [1] and individuals with disabilities, expected to be of special concern in the light of a declining caregiving workforce [43] and increased demand [10]. Joint attention on shared objects, a component of handover, has been shown to encourage social closeness [52]. Long-term impact of human-robot bonding on older adult isolation or childhood development remains understudied. Social robots developed without coupling short-range focused data such as ours with studies on longer-range attachment behavior are likely to introduce **negative societal consequences**. If a robot on which an individual has developed a dependence malfunctions or has to be discontinued, it may introduce anxiety and regression of collaborative goals within the individual, akin to the loss of a loved one. We share full details of our setup, capture, computing, and code in supplementary to promote continued wider-scale data collections for societally-aware HRI.

## Acknowledgments

This work was funded by National Science Foundation grant IIS-2023998. We thank Mingjun Li and Nikolas Lamb for insightful discussions on algorithm development and sensor calibration. We also thank Priyo Ranjan Kundu Prosun, Thomas Dubay, Ben Molloy, Irfan Yaqoob, Numan Zafar, Jyothinadh Minnekanti, Sichao Li, Houchao Gan, Xinchao Song, Kun Han, Alaina Tulskie, Holly Rossmann, Rosalina Delwiche, Cameron Hood, Odin Kohler, Alexander Cohen, Christian Soucy, and Gianna Voce for assistance with data collection and data annotation.

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
