# HOH: Markerless Multimodal Human-Object-Human Handover Dataset with Large Object Count Supplemental Documentation

**Noah Wiederhold**
Clarkson University
wiedern@clarkson.edu

**Ava Megyeri**
Clarkson University
megyeram@clarkson.edu

**DiMaggio Paris**
Clarkson University
parisda@clarkson.edu

**Sean Banerjee**
Clarkson University
sbanerje@clarkson.edu

**Natasha Kholgade Banerjee**
Clarkson University
nbanerje@clarkson.edu

## 1 Dataset Link and Password

A landing page for access to the data has been created as follows:

1. Dataset Page: https://hohdataset.github.io/
2. Dataset Access: email first author (see landing page)

## 2 Dataset Information

The HOH dataset contains multimodal data from a variety of cameras. This data has been processed to include skeletons, point clouds, and segmentation masks. A summary of the included data is provided in Figure 1, as well as section 3 in the main paper.

**Data Format**   All capture data is saved in 178 directories that represent the recording of multiple interactions. The number of interactions differs in every directory due to a 10-minute maximum recording time. Some recordings were stopped before 10 minutes had elapsed in order to redo a mistake made by the participants, e.g. the receiver accidentally picked up the object after the role swap. This resulted in some recordings having very few interactions. The naming format for these 178 directories is *<giver ID>-<receiver ID>-S<starting interaction number>*. For example, the directory for the first recording of giver 01638 and receiver 46157 is named "01638-46157-S1". In each of these 178 directories, there are 6 sub-directories:

1. *Azure* - Contains Azure Kinect color videos and Azure Kinect depth videos for each of the 4 cameras. Color videos are named [NUM].mp4, depth videos [NUM].mkv, and viewable depth videos as [NUM]_depth.mp4, where [NUM] represents a number from 0 to a maximum of 18, based upon the quantity of interactions that would have fit in the 10-minute duration.

2. *MaskTracking* - Contains a zip file with all tracked masks within the directory. The zip folder [RECORDING]_mask.zip contains directories for each of the 4 cameras. In each camera folder are all interaction object and hand mask files saved as Python NPZ format. The NPZ file contains a stacked numpy array of masks, one for each frame in the interaction. The zip folder [RECORDING]_maskcorrected.zip has a similar structure, though each NPZ file contains an individual mask for the frame that was fixed.

37th Conference on Neural Information Processing Systems (NeurIPS 2023) Track on Datasets and Benchmarks.

# HOH Dataset Facts

**Dataset** Human-Object-Human Handover

---

Motivation

---

**Summary** A markerless 3D multimodal dataset on human-human handovers with 20 participant pairs using 136 objects spanning 8 everyday-use classes

**Example Use Cases** Human-robot handover research, cognitive psychology research, exploration of automated human and object pose estimation algorithms

**Original Authors** N. Wiederhold, A. Megyeri, D. Paris, S. Banerjee, N. K. Banerjee

---

Metadata

---

**URL** [https://hohdataset.github.io](https://hohdataset.github.io)
**Released** October 27, 2023

---

Sensors

---

| | |
|---|---|
| **Azure Kinect Color** | 4 |
| **Azure Kinect Depth** | 4 |
| **FLIR PointGrey Blackfly S High Speed Color** | 4 |

Object Classes

| | |
|---|---|
| **Total Objects** | 136 |
| **Toys** | 19 |
| **Food/Drink** | 19 |
| **Cooking** | 24 |
| **Tool** | 15 |
| **Mug** | 12 |
| **Office** | 11 |
| **Household** | 36 |

---

Participants

---

| | |
|---|---|
| **Total participants** | 40 |
| **Total Pairs** | 20 |
| **M/M Pairs** | 16 |
| **M/F Pairs** | 4 |
| **Gender** | 34M, 6F |
| **Age** | 24.8±7.4 |

---

Data Size

---

| | |
|---|---|
| **Total Size** | 9.51 TB |
| **Azure** | 195.80 GB |
| **Mask Tracking** | 2.37 GB |
| **OpenPose** | 927.38 MB |
| **PCFiltered** | 94.34 GB |
| **PCFull** | 4.54 TB |
| **PointGrey** | 4.66 TB |
| **3D Model Alignments** | 26.7 MB |

Figure 1: A dataset informational card for HOH

3. *OpenPose* - Contains a zip file with all skeletons in JSON format, following the [NUM] notation, one [NUM] folder per interaction.

4. *PCFiltered* - Contains a zip file with all object and hand point clouds, following the [NUM] notation. Cleaned versions of the point clouds are also available in the *Cleaned* folder for each interaction.

5. *PCFull* - Contains a zip file with all full scene point clouds, following the [NUM] notation.

6. *PointGrey* - Contains a zip file with all Point Grey images.

7. *3dModelAlignments* - Contains a zip file with transformations that align the 3D model of the object used in each handover to the object in the scene point cloud for each timestep.

Within each directory, point clouds, videos, and masks are generated between the giver and receiver contact frames (G and R) inclusive, and where trackable for masks and segments. Other than the 178 data directories, there exist 4 directories, called *Objects*, *Code*, *Calibration*, and *ParticipantInfo*:

The *Objects* directory contains 3D models and metadata for all 136 objects. All of the 3D models are stored in a sub-directory called *3d_models*. Inside of *3d_models*, there is one directory corresponding to each object, named according to object ID. For all objects excluding 116 and 120, multiple 3D models are present which are discussed in Section 8.

The *Code* directory contains all code used to collect and process the data. The *Code* directory contains three sub-directories, named *Acquisition*, *Experiments*, and *Processing*. The *Acquisition* directory contains all code used for data acquisition, sorted into two sub-directories by language used: C# and Python. The *Experiments* directory contains all code used for the experiments described in Section 6, broken down into *Grasp*, *Orientation*, and *Trajectory* sub-directories. The *Processing* directory contains all code used for data processing. All code will be published on GitHub for public use upon acceptance, along with documentation in README files.

The *ParticipantInfo* directory contains the following:

1. *demographics_responses.csv* - The answers submitted by the participants for the demographics questionnaire.

2. *grasp_handedness.csv* - Grasp handedness labels for each interaction.

3. *grasp_taxonomy.csv* - Grasp taxonomy labels for each interaction.

4. *participant_seating.json* - Participant seating arrangements, organized by whether the giver is sitting on the left or right of the capture environment.

5. *Participant_Form_Responses* - A directory that contains all data collected from the digitization of the participant and experimenter forms, as detailed in Paragraph 3. The naming format inside this directory follows the convention: *<giver ID>-<receiver ID>.csv*.

The *Calibration* directory contains the following:

1. *group1_calib* - A directory containing calibration intrinsic and extrinsic parameters for sessions that use object set 1 as detailed in Section 4.

2. *group2_calib* - A directory containing calibration intrinsic and extrinsic parameters for sessions that use object set 2 as detailed in Section 4.

3. *fine_tuned_transforms* - A directory containing the fine-tuned transformations for all sessions as detailed in Section 4.

As shown by the data card in Figure 1, the dataset is nearly 9.51 TB, with the main high-file-size component being the full point clouds. Azure data is available in the form of videos. Though Point Grey data is currently shared as images, we plan to compress them into video files and expect the file sizes to be approximately twice the size of the Azure color videos, thereby greatly improving compression. Filtered point clouds for the objects and hands occupy a considerably reduced size on disk due to their small point count. We plan to provide down-sampled options for full point clouds.

Example 3D visualizations including full scene point clouds and isolated giver hand, object, and receiver hand point clouds, are shown in Figure 2.

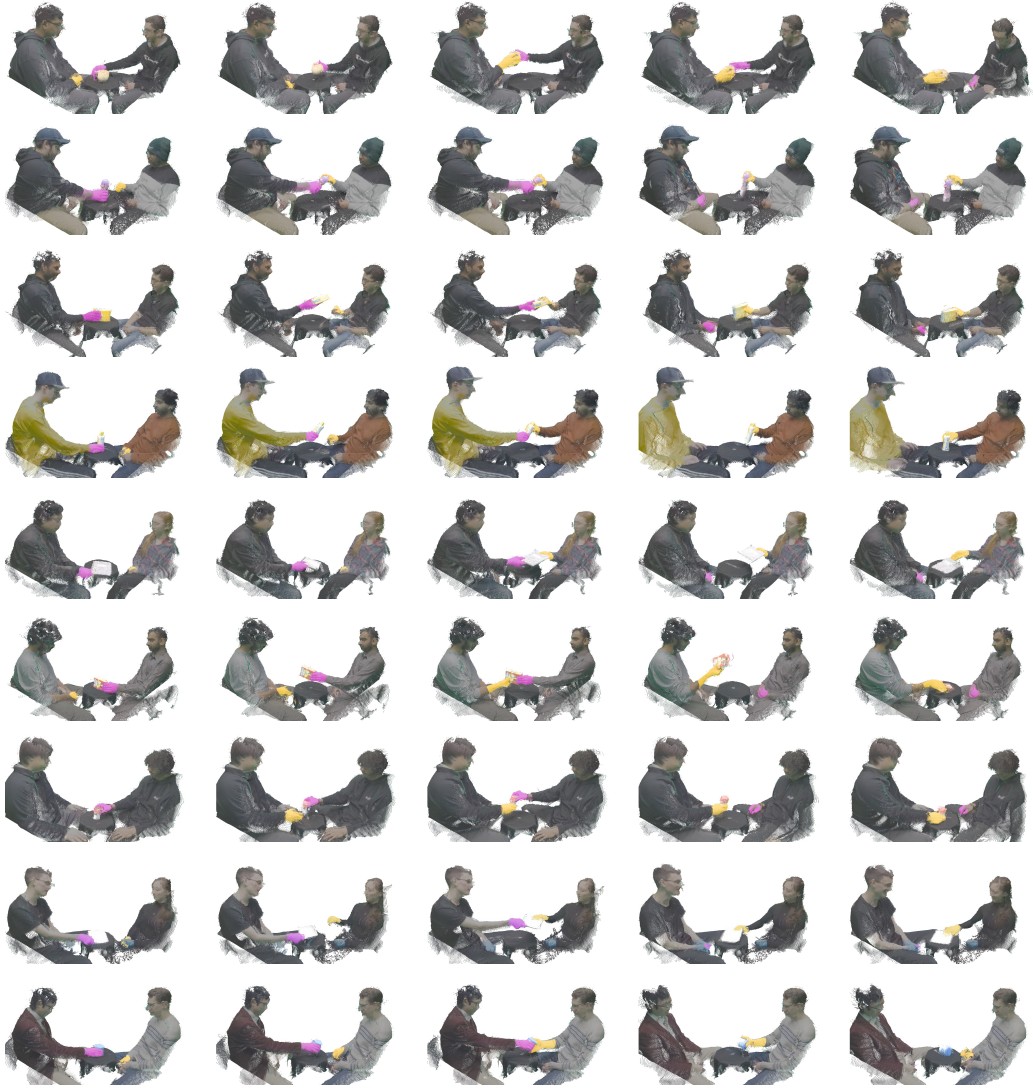

Figure 2: Example 3D visualizations of full scene point clouds at 5 time points during a handover interaction, with Frame G (point of first giver contact) in the leftmost column, Frame T (point of transfer) in the center column, and Frame R (point of last receiver contact) in the rightmost column. The giver hand is highlighted magenta and the receiver hand is highlighted gold.

**License Information.** We license all new assets in the dataset, including but not limited to the color and depth images, all versions of object models, manual annotations, all varieties of point clouds, segmentation masks, body skeletons, and participant demographic and comfort data, under the Creative Commons Attribution-NonCommercial 4.0 International (CC BY-NC 4.0) License (https://creativecommons.org/licenses/by-nc/4.0/), with the only exceptions being those object models in Table 1. We make use of 14 3D models from Thingiverse, for which licensing information is displayed in Table 1, and attribution information in Table 2. We do not release altered meshes for objects 116 and 120. We assign them an ID that is compliant with our naming and categorization scheme for objects. All code publicly released with this dataset, including code which allows for loading, modification, and application of the data, is licensed under the MIT License (https://opensource.org/license/mit/).

Table 1: Licensing information for object models acquired from Thingiverse.

| Object ID | License |
|-----------|---------|
| 115 | Creative Commons - Attribution License |
| 116 | Creative Commons - Attribution - Non-Commercial - No Derivatives License |
| 118 | Creative Commons - Attribution License |
| 120 | Creative Commons - Attribution - Non-Commercial - No Derivatives License |
| 121 | Creative Commons - Attribution License |
| 122 | Creative Commons - Attribution - Non-Commercial - Share Alike License |
| 127 | Creative Commons - Attribution - Non-Commercial License |
| 128 | Creative Commons - Public Domain Dedication License |
| 129 | Creative Commons - Attribution License |
| 221 | Creative Commons - Attribution - Share Alike License |
| 233 | Creative Commons - Attribution - Non-Commercial - Share Alike License |
| 234 | Creative Commons - Attribution License |
| 235 | Creative Commons - Attribution - Non-Commercial License |
| 236 | Creative Commons - Attribution License |

Table 2: Licensing attribution for object models acquired from Thingiverse.

| Object ID | Thingiverse Author | Link To Thingiverse Page |
|-----------|--------------------|--------------------------|
| 115 | @RandomUser23447274 | https://www.thingiverse.com/thing:4694553 |
| 116 | @Clms31 | https://www.thingiverse.com/thing:4690097 |
| 118 | @ertugrulozarozar | https://www.thingiverse.com/thing:4715797 |
| 120 | @MarVin_Miniatures | https://www.thingiverse.com/thing:4038181 |
| 121 | @bert_lz | https://www.thingiverse.com/thing:4688251 |
| 122 | @stratosvasilas | https://www.thingiverse.com/thing:4694905 |
| 127 | @riskable | https://www.thingiverse.com/thing:2173745 |
| 129 | @Cool3DModel | https://www.thingiverse.com/thing:2445539 |
| 221 | @david4974 | https://www.thingiverse.com/thing:1617958 |
| 233 | @sffubs | https://www.thingiverse.com/thing:4684367 |
| 234 | @tobymerritt | https://www.thingiverse.com/thing:4695393 |
| 235 | @Onil_Creations | https://www.thingiverse.com/thing:4700386 |
| 236 | @clanmcfadden | https://www.thingiverse.com/thing:4688105 |

**Author Statement of Responsibility.** The authors confirm all responsibility in case of violation of rights and confirm the license associated with the dataset and code.

**Dataset Accessibility and Long-Term Preservation Plan.** Upon acceptance, we plan to host the full dataset on our local datacenter and make it available through the project webpage. We plan to host a compressed version of the dataset on Google Drive associated with our institution. Code will be hosted on GitHub. The project page will be hosted on GitHub to ensure that the data remains accessible.

**Dataset Identifier.** Access the persistent landing page for the dataset here: https://tinyurl.com/hohdataset

## 2.1 Datasheets For Datasets

We follow the framework of Datasheets for Datasets [5] for our dataset documentation and intended uses.

1. **Motivation**

   (a) **For what purpose was the dataset created?**
   To accelerate data-driven research on handover studies, human-robot handover implementation, and artificial intelligence on handover parameter estimation from reality-representative 2D and 3D data of natural person interactions.

   (b) **Who created the dataset (e.g., which team, research group) and on behalf of which entity (e.g., company, institution, organization)?**
   Terascale All-sensing Research Studio at Clarkson University.

   (c) **Who funded the creation of the dataset?**
   This work was funded by National Science Foundation grant IIS-2023998.

2. **Composition**

   (a) **What do the instances that comprise the dataset represent (e.g., documents, photos, people, countries)?**
   The dataset is comprised of images (.png, .jpg), videos (.mp4,mkv), texture-mapped point clouds (.ply), segmentation masks (.npz), skeletons (.json), object models (.obj), and spreadsheets (.csv).

   (b) **How many instances are there in total (of each type, if appropriate)?**
   See Figure 1 and main paper Table 2 for detailed breakdown.

   (c) **Does the dataset contain all possible instances or is it a sample (not necessarily random) of instances from a larger set?**
   The dataset contains all possible instances.

   (d) **What data does each instance consist of?**
   Each data instance is a single handover interaction, including color and depth images, point clouds, segmentations, skeletons, and annotations.

   (e) **Is there a label or target associated with each instance?**
   Each handover has an object 3D model associated with it. Also, each handover is linked to rich metadata including participant comfort ratings, object metadata, participant demographics information, and experimenter notes. Each handover is named according to the ID numbers of the participants involved and a serial number denoting the place of the interaction in the overall sequence.

   (f) **Is any information missing from individual instances?**
   A few Azure Kinect images were dropped during data collection. Full scene point clouds, skeletons, and masks are missing for these dropped frames.

   (g) **Are relationships between individual instances made explicit (e.g., users' movie ratings, social network links)?**
   All instances are explicitly grouped by the participant dyad involved.

   (h) **Are there recommended data splits (e.g., training, development/validation, testing)?**
   Not at this time. Users of this dataset are encouraged to experiment and divide the dataset as it suits their applications.

   (i) **Are there any errors, sources of noise, or redundancies in the dataset?**
   An extraction error occurred that affects <1% of the Point Grey color images. The depth data can be somewhat noisy when recording from as far away as the depth sensors are in our system. This is mitigated through 2D segmentation and 3D-based noise and outlier removal. No redundancies exist based on the knowledge of the authors.

   (j) **Is the dataset self-contained, or does it link to or otherwise rely on external resources (e.g., websites, tweets, other datasets)?**
   The dataset is self-contained, though it includes 14 other 3D models for objects. See Table 1 for details.

   (k) **Does the dataset contain data that might be considered confidential (e.g., data that is protected by legal privilege or by doctor patient confidentiality, data that**

**includes the content of individuals' non-public communications)?**
No.

(l) **Does the dataset contain data that, if viewed directly, might be offensive, insulting, threatening, or might otherwise cause anxiety?**
No.

(m) **Does the dataset identify any subpopulations (e.g., by age, gender)?**
This information is present in the demographics responses from each participant, though the dataset is not subdivided corresponding to any of the demographics data.

(n) **Is it possible to identify individuals, either directly or indirectly (i.e., in combination with other data) from the dataset?**
Yes. The dataset includes color data depicting all participants who consented to the public release of their color data.

(o) **Does the dataset contain data that might be considered sensitive in any way (e.g., data that reveals race or ethnic origins, sexual orientations, religious beliefs, political opinions or union memberships, or locations; financial or health data; biometric or genetic data; forms of government identification, such as social security numbers; criminal history)?**
No.

3. **Collection Process**

(a) **How was the data associated with each instance acquired?**
Images were collected using 4 Microsoft Azure Kinect cameras and 4 FLIR Point Grey Blackfly S cameras. See Paragraph 4 for details about the sensors, and see Section 3 in the main paper for procedure details. 3D models were scanned using an Einscan-SP 3D-scanner, as detailed in Paragraph 8.

(b) **What mechanisms or procedures were used to collect the data (e.g., hardware apparatuses or sensors, manual human curation, software programs, software APIs)?**
For detail about the sensors used, see Section 4. For detail about the data collection procedure, see Section 3 in the main paper. All code to control sensors and manipulate data was written internally, excluding the Azure Kinect API and Spinnaker API which are used to control sensors.

(c) **Who was involved in the data collection process and how were they compensated)?**
Students were recruited to administrate the data collection sessions. The student experimenters were compensated with course credit.

(d) **Over what timeframe was the data collected?**
The data was collected between February 24, 2023 and April 5, 2023.

(e) **Were any ethical review processes conducted (e.g., by an institutional review board)?** The project received approval from the institutional review board prior to data collection.

(f) **Did you collect the data from the individuals in question directly, or obtain it via third parties or other sources?**
Data was collected from the participants directly.

(g) **Were the individuals in question notified about the data collection?**
Yes. Participants were recruited voluntarily. The message used to recruit the participants can be found in Paragraph 3. The speech read to participants that details the data collected can be found in Paragraph 3.

(h) **Did the individuals in question consent to the collection and use of their data?**
Yes, each participant signed an informed consent document where they consented to being videotaped and allowed the release of their color data and non-identifiable data. Two participants in this study did not consent to having their color data publicly released, and in these cases we withhold any of their identifiable data and release their non-identifiable data.

(i) **If consent was obtained, were the consenting individuals provided with a mechanism to revoke their consent in the future or for certain uses?**
Participants were reminded on multiple occasions that they may stop the study at any point if they wish. It was clarified that a participant may ask for their data to be

deleted at any time, even after the conclusion of the study, by communicating with an experimenter and mentioning their 5-digit participant ID number.

(j) **Has an analysis of the potential impact of the dataset and its use on data subjects (e.g., a data protection impact analysis) been conducted?**
Yes, for instance, a comprehensive assessment of risks has been performed and communicated to the participants via the informed consent form. Subjects also had the opportunity of requesting that identifiable color not be released via the informed consent form. Our dataset does not release color data for two participant pairs based on one participant in each pair having opted out of identifiable color data release.

4. **Preprocessing, Cleaning and Labelling**

(a) **Was any preprocessing/cleaning/labeling of the data done (e.g., discretization or bucketing, tokenization, part-of-speech tagging, SIFT feature extraction, removal of instances, processing of missing values)?**
Yes, we performed annotation and ran software on the images. See question 4(c).

(b) **Was the "raw" data saved in addition to the preprocessed/cleaned/labeled data (e.g., to support unanticipated future uses)?**
Yes. The raw data is saved separately as videos and images.

(c) **Is the software that was used to preprocess/clean/label the data available?**
Yes. SAM [6], OpenPose [2], and Track Anything [10] are publicly available.

5. **Uses**

(a) **Has the dataset been used for any tasks already?**
No.

(b) **Is there a repository that links to any or all papers or systems that use the dataset?**
As the dataset has not been publicly released yet, there are no papers that use the dataset at present.

(c) **What (other) tasks could the dataset be used for?**
The dataset could be used for training assistive robots for purposes such as in-home care for the elderly or providing help in the kitchen by retrieving a utensil or an ingredient when a person may not have the free hands or the time to do it on their own.

(d) **Is there anything about the composition of the dataset or the way it was collected and preprocessed/cleaned/labeled that might impact future uses?**
The use of entirely non-invasive, markerless data collection techniques could impact the ability to obtain dense ground truth data, e.g. object pose, without substantial manual effort.

(e) **Are there tasks for which the dataset should not be used?**
This dataset should not be used to cause a robot to intentionally give an object unsafely, e.g. extend the blade of a knife directly toward a human user.

6. **Distribution**

(a) **Will the dataset be distributed to third parties outside of the entity on behalf of which the dataset was created?**
Yes. The dataset will be made publicly available upon acceptance.

(b) **How will the dataset will be distributed (e.g., tarball on website, API, GitHub)?**
The dataset will be hosted on our local datacenter, and potentially Google Drive and GitHub. See Paragraph 2 for further details.

(c) **When will the dataset be distributed?**
Upon acceptance.

(d) **Will the dataset be distributed under a copyright or other intellectual property (IP) license, and/or under applicable terms of use (ToU)?**
Yes, see Paragraph 2 for details.

(e) **Have any third parties imposed IP-based or other restrictions on the data associated with the instances?**
No.

(f) **Do any export controls or other regulatory restrictions apply to the dataset or to individual instances?**
No.

7. **Maintenance**

    (a) **Who will be supporting/hosting/maintaining the dataset?**
    The authors of this work will be hosting and maintaining the dataset.

    (b) **How can the owner/curator/manager of the dataset be contacted (e.g., email address)?**
    All authors can be contacted through the email addresses listed on the first page of the paper.

    (c) **Is there an erratum?**
    No.

    (d) **Will the dataset be updated (e.g., to correct labeling errors, add new instances, delete instances)?**
    The dataset is likely to be expanded in the future with more ground truth and to be tailored to specific applications, e.g. 2D background replacement for more effective deep learning.

    (e) **Are there applicable limits on the retention of the data associated with the instances (e.g., were the individuals in question told that their data would be retained for a fixed period of time and then deleted)?**
    No limits have been placed on the data. The only information provided to participants is that they can choose to opt out of public release of identifiable color information.

    (f) **Will older versions of the dataset continue to be supported/hosted/maintained?**
    Older versions of the dataset will only be expanded upon, not entirely replaced.

    (g) **If others want to extend/augment/build on/contribute to the dataset, is there a mechanism for them to do so?**
    Not currently, as the dataset is large and must be hosted on our private datacenter at present. Collaboration may be possible in the future with substantial compression.

## 3   Participant Forms and Messages

In this section, we provide further detail for contact with participants and participant form responses.

**Participant Recruitment Message.** Participants were recruited voluntarily via communication at the institution where the study was conducted. The recruitment message was posted on online collaborative work spaces (* denotes information that is redacted to preserve anonymity):

```
Subject: Participants sought for research study on understanding human
preferences for handover parameters for safe human-robot collaboration

You are receiving this request as part of a *-wide announcement on
recruitment for this study.  We seek participants for a research study
on understanding human preferences for handover parameters for safe
human-robot collaboration.  We are looking for individuals aged 18 to 99,
with no known upper limb disability or injury that interferes with curling,
grasping, and lifting, and that have no injury to fingers on either hand.

The data collection will take no longer than 2 hours.  Participants will
complete a demographics questionnaire and take part in a set of experiments
involving interacting with 68 objects.

Participants will be recorded using Azure Kinect cameras and Point Grey
Blackfly S cameras to gather data on human body posture and contact regions
on objects.  The study will enable us to design algorithms for robots that
are aware of human handover preferences, so as to ensure safe human-robot
collaboration.

The safety of all participants in this study is of paramount importance.
We request that all subjects take a COVID-19 test within the 3 days prior
to their data collection session.  If you are unable to take a test within
this window, we will provide you with one.

To remain in compliance with CDC, state, and institutional safety
regulations for COVID-19, participants should not have left the * County
area up to 14 days prior to your session.  Participants will be recruited
```

```
if they do not exhibit the following symptoms and have not exhibited them
for 14 days:  fever or chills, cough, shortness of breath or difficulty
breathing, fatigue, muscle or body aches, headache, new loss of taste
or smell, sore throat, congestion or runny nose, nausea or vomiting, and
diarrhea.

To reach us for participation in this study, please email * * at *@*.*.  If
you wish to opt-out of follow-up emails, please respond with a note stating
that you do not want to receive future emails about participating in the
study.  The * IRB approval number for this study is * and the contact
information for the * IRB office is * via email, and (***) ***-**** via
phone.
```

**Participant Arrival Message.**   Upon arrival for their data collection session, participants were read the following by an experimenter:

*Hello, my name is (research personnel) and we are conducting a research study on understanding human preferences for handover parameters for safe human-robot collaboration.*

*The purpose of this study is to understand the relationship between object form and function, and human preferences for handover parameters such as where you hold an object, what orientations and distances you prefer an object being handed to you, and at what point do you prefer that an object be released upon handover. Our study will help to design robots that are aware of human handover preferences, to ensure safe human-robot collaboration in home and work environments, for example, safe assistive robots to help older adults.*

*Today you will first read the informed consent form, and then take part in an experiment where you will lift these 68 objects (shown), one by one, and hand them to your partner. At the start, one of you will be assigned the role of hander and the other will be assigned the role of receiver. The hander will be asked to give the object to the receiver, and then you both will fill out a response about the handover on these forms (clipboard). Please stow the clipboards next to your chairs while the capture is taking place. Half of the way through the session we'll ask you to switch the hander and receiver roles. For some of the handover interactions, you may be asked to wear blue nitrile gloves.*

*As mentioned, you will follow each handover interaction with a response on the paper forms. With this response you are to indicate your level of comfort with the interaction that just occurred on a scale of 1 ("not comfortable at all") to 7 ("the most comfortable - a perfect handover"). You can think of a rating of 1 as representing a handover that was barely complete, where you may be forced to use an uncomfortable grasp, the timing is off, and/or the location or orientation of the object is not preferable. A comfort rating of a 7 should represent a handover that cannot be improved; everything was done naturally in your opinion.*

*While handing objects back and forth, you will be recorded at all times using these 4 Azure Kinect depth sensors and Point Grey Blackfly S high speed color cameras that allow us to capture information about your body and hand skeleton while you perform the grasp and allow us to understand where and how you hold objects. The maximum weight of any object is no more than 8 lb which is about the same as a gallon of milk. Most objects are no more than 2-3 lb in weight, and they are all objects you may use in your home or office.*

*Please remember not to squeeze any object too hard because some of the objects are fragile. Just try to grasp the objects naturally. We also ask that you cover over clothing that is similar in color to the blue gloves or the green curtains (if necessary, we have neutral-colored shirts available). Finally, try to avoid bumping the camera frame or moving the chairs or table during captures. We'll take a break half way through when we switch roles so you can stand up and stretch a little.*

*For your safety, we are asking that you adhere to all safety and distancing regulations to minimize contamination.*

*Your name will be removed from our dataset so as to not be associated with your data. From this point forward, all of your data will be identified by your 5-digit ID number.*

Subject ID [________] as GIVER

Below, please indicate your level of comfort with each of the 68 interactions on a scale of 1 ("not comfortable at all") to 7 ("the most comfortable - a perfect handover"):

| Group 1 | Serial No. | 1 | 2 | 3 | 4 | 5 | 6 | 7 | 8 | 9 | 10 | 11 | 12 | 13 | 14 | 15 | 16 | 17 |
|---|---|---|---|---|---|---|---|---|---|---|---|---|---|---|---|---|---|---|
| Rating | 1 | | | | | | | | | | | | | | | | | |
| | 2 | | | | | | | | | | | | | | | | | |
| | 3 | | | | | | | | | | | | | | | | | |
| | 4 | | | | | | | | | | | | | | | | | |
| | 5 | | | | | | | | | | | | | | | | | |
| | 6 | | | | | | | | | | | | | | | | | |
| | 7 | | | | | | | | | | | | | | | | | |

| Group 2 | Serial No. | 18 | 19 | 20 | 21 | 22 | 23 | 24 | 25 | 26 | 27 | 28 | 29 | 30 | 31 | 32 | 33 | 34 |
|---|---|---|---|---|---|---|---|---|---|---|---|---|---|---|---|---|---|---|
| Rating | 1 | | | | | | | | | | | | | | | | | |
| | 2 | | | | | | | | | | | | | | | | | |
| | 3 | | | | | | | | | | | | | | | | | |
| | 4 | | | | | | | | | | | | | | | | | |
| | 5 | | | | | | | | | | | | | | | | | |
| | 6 | | | | | | | | | | | | | | | | | |
| | 7 | | | | | | | | | | | | | | | | | |

| Group 3 | Serial No. | 35 | 36 | 37 | 38 | 39 | 40 | 41 | 42 | 43 | 44 | 45 | 46 | 47 | 48 | 49 | 50 | 51 |
|---|---|---|---|---|---|---|---|---|---|---|---|---|---|---|---|---|---|---|
| Rating | 1 | | | | | | | | | | | | | | | | | |
| | 2 | | | | | | | | | | | | | | | | | |
| | 3 | | | | | | | | | | | | | | | | | |
| | 4 | | | | | | | | | | | | | | | | | |
| | 5 | | | | | | | | | | | | | | | | | |
| | 6 | | | | | | | | | | | | | | | | | |
| | 7 | | | | | | | | | | | | | | | | | |

| Group 4 | Serial No. | 52 | 53 | 54 | 55 | 56 | 57 | 58 | 59 | 60 | 61 | 62 | 63 | 64 | 65 | 66 | 67 | 68 |
|---|---|---|---|---|---|---|---|---|---|---|---|---|---|---|---|---|---|---|
| Rating | 1 | | | | | | | | | | | | | | | | | |
| | 2 | | | | | | | | | | | | | | | | | |
| | 3 | | | | | | | | | | | | | | | | | |
| | 4 | | | | | | | | | | | | | | | | | |
| | 5 | | | | | | | | | | | | | | | | | |
| | 6 | | | | | | | | | | | | | | | | | |
| | 7 | | | | | | | | | | | | | | | | | |

Figure 3: Template of the form completed by participants during a data collection session. The header changes depending on whether a participant is given the role of GIVER or RECEIVER.

*Please hold onto this card for the remainder of the study or in case you would like to contact us in the future. If at any point you feel any discomfort and wish to stop the study, please let me know, and we will stop the study immediately. You have the right to opt out of this study at any time you choose, and your data will immediately be erased.*

*If you have any questions, I am happy to answer them now.*

**Participant Paper Forms.** Participants completed the form in Figure 3 to provide feedback on how comfortable they felt with each interaction. After every handover interaction, the participant marks their level of comfort in the column that corresponds to the interaction that happened immediately prior. The column in each group labeled with the numbers 1-7 displays the rating represented by each row, and the participant marks a single box in each column to provide their level of comfort with each interaction.

**Experimenter Forms.** Experimenters completed the form in Figure 4 during the data collection session. The giver and receiver IDs are recorded along with their seating position. Since objects were randomly selected, experimenters recorded the object ID used for each interaction. The serial number columns represent the interaction index.

**Paper Form Digitization Template.** All information written in the experimenter and participant paper forms is digitally entered into the form shown in Figure 5 by experimenters after each data collection session. To ensure the quality of the digitization by experimenters, code was used to validate the forms, which inspected object IDs, glove count, giver and receiver IDs, and comfort ratings.

| GiverID: | | ReceiverID: | | | | | |
|---|---|---|---|---|---|---|---|
| right or left: | | right or left: | | | | | |
| | | | | | | | |
| Serial no. | Object ID | Serial no. | Object ID | Serial no. | Object ID | Serial no. | Object ID |
| 1 | | 18 | | 35 | | 52 | |
| 2 | | 19 | | 36 | | 53 | |
| 3 | | 20 | | 37 | | 54 | |
| 4 | | 21 | | 38 | | 55 | |
| 5 | | 22 | | 39 | | 56 | |
| 6 | | 23 | | 40 | | 57 | |
| 7 | | 24 | | 41 | | 58 | |
| 8 | | 25 | | 42 | | 59 | |
| 9 | | 26 | | 43 | | 60 | |
| 10 | | 27 | | 44 | | 61 | |
| 11 | | 28 | | 45 | | 62 | |
| 12 | | 29 | | 46 | | 63 | |
| 13 | | 30 | | 47 | | 64 | |
| 14 | | 31 | | 48 | | 65 | |
| 15 | | 32 | | 49 | | 66 | |
| 16 | | 33 | | 50 | | 67 | |
| 17 | | 34 | | 51 | | 68 | |

Figure 4: Template of the form completed by experimenters during a data collection session.

## 4 Data Capture System

In this section we provide additional detail about the capture system used for HOH.

**Camera setup.** Our data capture setup consists of a 1.7m × 1.7m × 2.0m T-slot frame rig with 4 Microsoft Azure Kinect RGB-D sensors and 4 FLIR Point Grey Blackfly S high-speed color cameras. The Kinect cameras are configured such that they record color (1920x1080 pixels) and depth (640x576 pixels) images at 30 frames per second (FPS). The Point Grey cameras have a 2.8-10 millimeter lens and record 60 FPS color (1440x1080 pixels) images. All extracted color images are stored as .jpg files. All extracted depth images are stored as 16-bit .png files. Kinects and Point Grey cameras are rigidly mounted on each of the corners of the capture system using custom-fabricated 3D-printed mounts at about 4 feet above the ground, directly pointed at the table. 1 Kinect/Point Grey pair is mounted on each vertical edge of the frame, 2 pairs at the front of the rig and 2 at the back as shown in Figure 6. A table and two opposing chairs are located in the capture environment, and cameras are pointed to enable full capture of the handover space and the face, hands, and body posture of both participants. The Kinect depth sensors perform optimally with a target that is between 1.5 and 4 feet away from the sensor, and are mounted on the capture system corners to ensure that the center of the table is in the middle of that range. The Point Grey cameras have no such restriction, and are mounted above the Kinects on the corner columns, pointing directly at the table and configured to have as much of the scene in focus as possible. A Dragon Touch camera is mounted at the top of the system to provide a live video feed to the experimenters outside the capture space, which is not recorded, and is shown in Figure 7. All Kinects are administrated by a high-performance computer which commands the cameras and coordinates all recorded data and transfers it to the long term storage. All Point Grey cameras are connected to another similar computer with the same purpose. This configuration of sensors, control computers, and hardware has been tested over varied recording lengths of up to 15 minutes, and consistently yields frame drop amounts below 0.2%.

| | giver id | right/left | mask? | receiver id | right/left | mask? | |
| | object id | giver response | receiver response | gloved? | bimanual? | stopped here? | notes |
|---|---|---|---|---|---|---|---|
| 1 | | | | | | | |
| 2 | | | | | | | |
| 3 | | | | | | | |
| 4 | | | | | | | |
| 5 | | | | | | | |
| 6 | | | | | | | |
| 7 | | | | | | | |
| 8 | | | | | | | |
| 9 | | | | | | | |
| 10 | | | | | | | |
| 11 | | | | | | | |
| 12 | | | | | | | |
| 13 | | | | | | | |
| 14 | | | | | | | |
| 15 | | | | | | | |
| 16 | | | | | | | |
| 17 | | | | | | | |
| 18 | | | | | | | |
| 19 | | | | | | | |
| 20 | | | | | | | |
| 21 | | | | | | | |
| 22 | | | | | | | |
| 23 | | | | | | | |
| 24 | | | | | | | |
| 25 | | | | | | | |
| 26 | | | | | | | |
| 27 | | | | | | | |
| 28 | | | | | | | |
| 29 | | | | | | | |
| 30 | | | | | | | |

Figure 5: Template of the form completed by experimenters after a data collection session. Note that the form is truncated in length for display purposes, but the actual form extends to serial index 68.

**Networking.** The Kinect control computer is comprised of an Intel Core i9-9900k 8-core CPU and a GeForce RTX 2060 Super GPU in order to effectively support 4 Kinects recording simultaneously, and 2 NVMe and 3 solid state hard drives to enable writing of the large volume of recorded data. The Point Grey control computer uses a Ryzen Threadripper 1900x 8-core CPU and GeForce GTX 1080ti GPU, as the Point Grey sensors do not utilize the GPU. The Point Grey control computer also makes use of 3 NVMe and 2 solid state hard drives to accommodate the data writing in real time. Kinects and Point Grey cameras are connected via USB 3.0 to their control computers. Both sensor control computers make use of a separate PCIe USB expansion unit to allow for distributed use of motherboard lanes when processing data. Both sensor control computers are connected to an administrative DHCP server, hosted on a 576TB network-attached storage device (NAS). The NAS facilitates communication between computers and serves as long-term storage for data collected by the system. Connections to the NAS are 10-Gigabit in order to alleviate data transfer bottlenecks. Figure 8 illustrates the inter-device connections that form the capture system network.

**Data Synchronization.** Color, IR, and depth frames from the same Kinect are hardware synchronized, though frames from different devices are not synchronized to each other. We use the flash of a bright green light in each recording to identify a time step in the color images. This light flash is automatically detected using an image differencing technique that examines computes the average green intensity of a specific region of the images. This technique automatically detects the significant inter-frame green intensity change when the light deactivates. The frame where the light turns off for the last time is automatically detected on each color camera, and this frame is considered to be the first frame of the recording. The frame index of the new first frame is propagated to the depth and IR image streams from each Kinect color image stream. Synchronization was validated for every recording through manual viewing of an image tile containing images from each camera that show the 3-image sequence before, during, and after the synchronization light turned off. If the

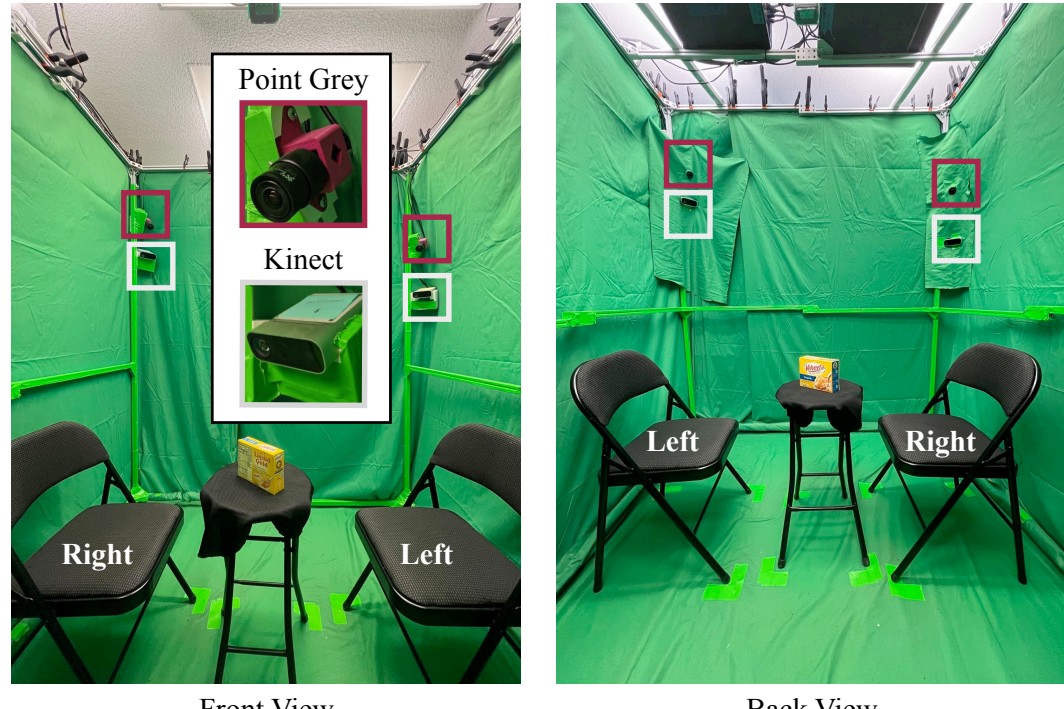

Front View                              Back View

Figure 6: Capture setup showing placement of cameras and seating.

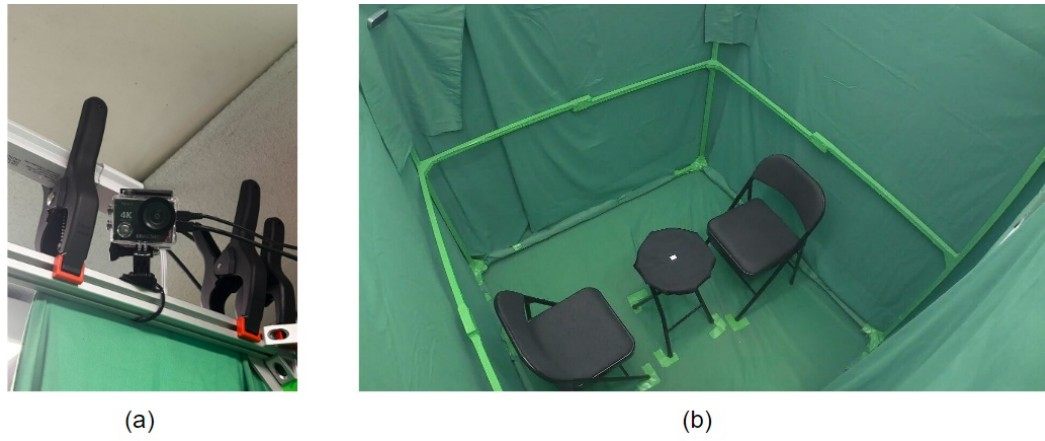

(a)                                      (b)

Figure 7: (a) DragonTouch camera that provides (b) a live feed into the capture environment.

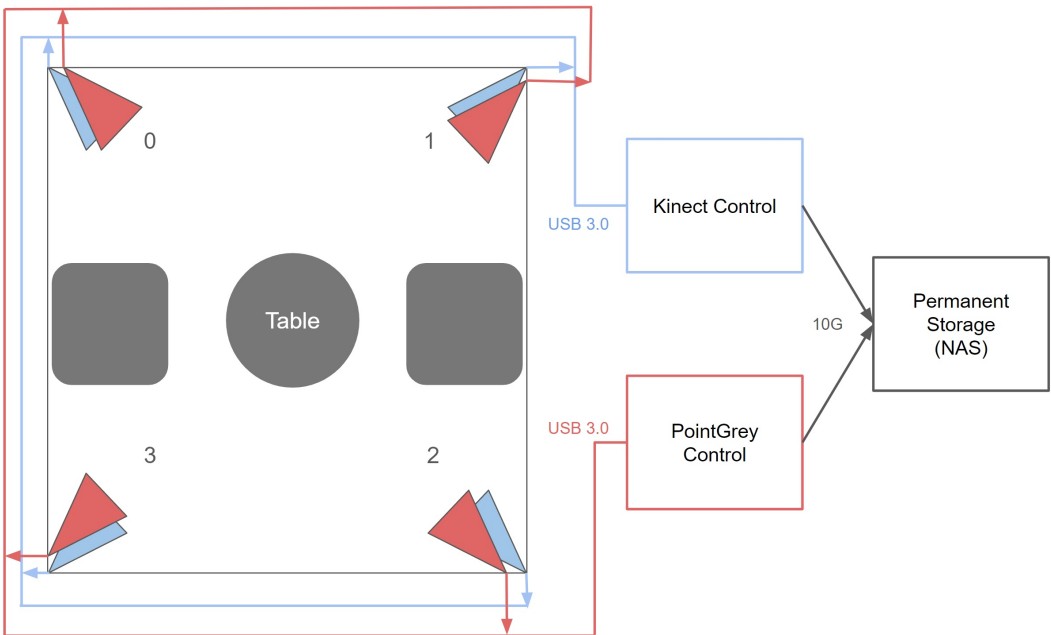

Figure 8: Networking diagram with Point Grey cameras in red and Kinect cameras in blue. Both camera types are connected to the respective control computers via USB 3.0. The control computers are connected to the NAS, where all data is stored.

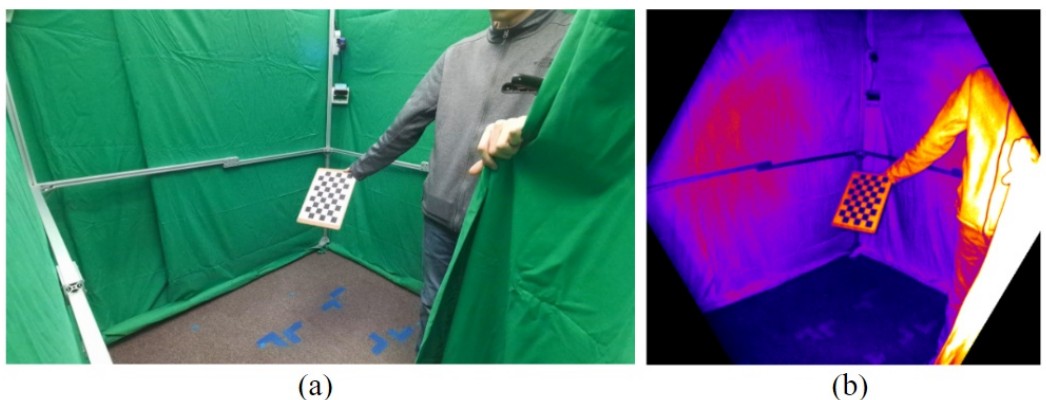

Figure 9: View of the calibration from the corner 0 (a) Kinect color and (b) Kinect infrared sensors.

image tile was not strictly in this format, which happened in less than 15% of cases, the synchronization offsets were manually corrected by adjusting the index of the new first frame to the point where the light actually shut off for each color image stream.

**Sensor Calibration.** We use checkerboard camera calibration similar to the method of Zhang [12] to obtain sensor intrinsic parameters and extrinsic parameters that relate the poses of pairs of sensors. The checkerboard calibration target used was a black and white 6x9 checkerboard printed on paper. The target was fixed to a rigid back plate to ensure that it would always remain flat. The squares have a side length of 30 millimeters. There is significant contrast between the white and black squares on the calibration target visible in color and Kinect IR images. The Kinect IR images are used to calibrate the Kinect depth sensor. We calibrate each color sensor to the Kinect depth sensor on each corner. We calibrate pairs of depth sensors together to traverse between the capture system corners. This scheme allows for transformation of data from any sensor to any other sensor in the capture system through a composition of at most 3 extrinsic transformations. In each calibration operation, one person moves within the capture system while constantly adjusting the pose of the

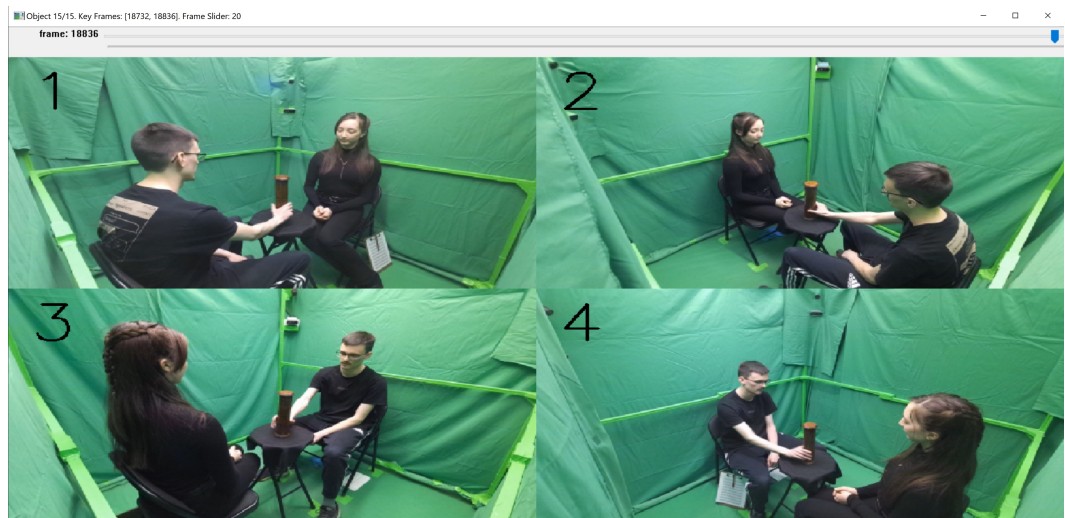

Figure 10: The annotation tool for selecting key events. Inputs are keypresses that correspond to temporal scrolling, frame selection, and saving.

checkerboard in a manner that the sensor(s) being calibrated can see such that the checkerboard spans the entire field of view and space of poses that are possible within the camera view. This process must be completed thoroughly in order to achieve an accurate estimation of sensor intrinsic and extrinsic parameters. The intrinsic and extrinsic parameters were computed using the Stereo Camera Calibrator app in MATLAB [7].

The data collection phase was split into two major participant groups, each interacting with a distinct set of 68 objects. Sensor calibration was performed prior to the recording of each of these groups, as this was a natural break in the recording process where calibration was possible. Calibration was performed multiple times to mitigate error from sensor drift over time. Each full-system calibration involved intrinsic parameter estimation for each individual sensor and extrinsic parameter estimation from multi-sensor calibration between pairs of sensors according to the above scheme. Over 200 images were used for each intrinsic calibration and over 1,000 were used for each extrinsic calibration in order to ensure less than 0.5 mean pixel error.

To obtain optimal point cloud scene reconstructions, we use Kinect depth sensor 0 as the reference coordinate frame and manually fine-tuned the extrinsic parameters to accurately transform the data from any depth sensor coordinate frame to the reference coordinate frame. This was done by applying the extrinsic transformations obtained from stereo calibration to transform the backprojected point clouds of depth sensor 1, 2, and 3 to the coordinate frame of depth sensor 0 of a single frame in one interaction in both groups. Using a tool designed with the vedo library [4] in Python, we manually rotated and translated the point clouds, aided by the Iterative Closest Point algorithm, to fine-tune the alignment with the reference coordinate point cloud. These fine-tuning transformations were saved and used when fusing the point clouds.

## 5   Additional Details for Ground Truth Annotation and Data Processing

In this section we provide additional details concerning the manual annotation processes and custom tools to obtain ground truth data originally mentioned in section 3 of the main paper. We also provide additional details concerning the implementation and execution of the post-processing mask tracking task introduced in section 3 of the main paper.

**Key Event Selection.**    Annotators used a tool created in-house to manually isolate 3 time points, referred to as key events, in each interaction that represent: first giver contact marking the grab portion of reach and grab phase called frame G, simultaneous giver and receiver grasp marking the middle region of the object transfer phase called frame T, and last receiver contact on the object marking the final part of the end of handover phase called frame R. The annotation tool, shown in 10,

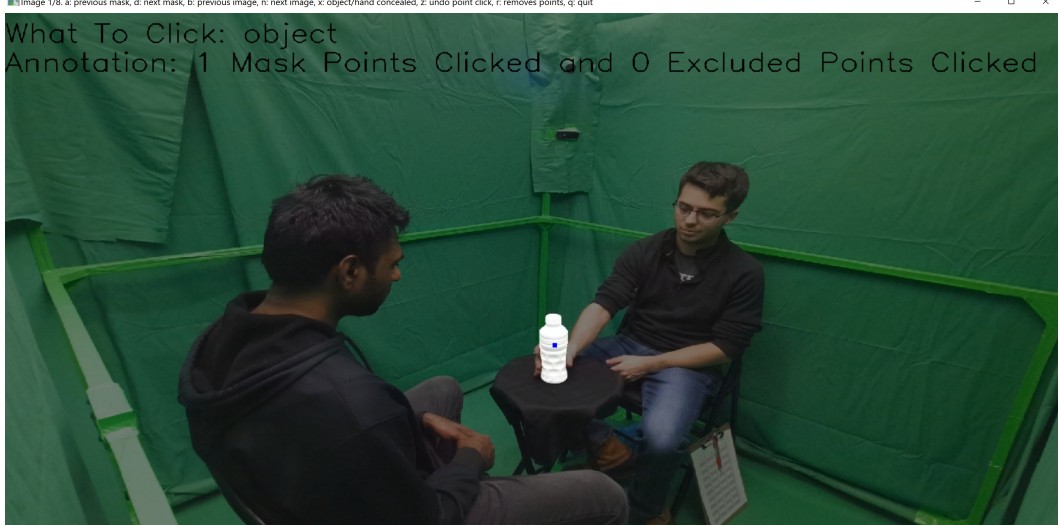

Figure 11: The annotation tool for selecting masks from the full image segmentation. Inputs are mouse clicks and keypresses which correspond to mask selection and saving, respectively.

uses the OpenCV [1] Python library to show the images and receive user input. The annotator selects a key event using keypresses, and they can scroll temporally through all images in the recording to select key events for all interactions. Once all key events are chosen, they are saved in .json files.

**Segmentation Mask Selection.** For each key event for all 4 Kinect color cameras for each interaction, annotators use a custom tool, shown in 11 to select 3 masks from the set of approximately 60 scene segmentation masks: that of giver's hand, the object, and the receiver's hand. Note that the receiver's hand is not present in frame G and the giver's hand is not present in frame R, and all 3 targets are present in frame T. Using this tool, the annotator clicks on the giver hand, object, or receiver hand in the image to select the corresponding mask, which is highlighted by the program. If the mask is correct, the annotator continues on to select the next mask in the process. If the mask is incorrect, the annotator clicks more foreground and background points, using left and right clicking respectively, to refine the clicked mask. If no background points are marked, the clicked masks are considered correct. If both foreground and background points are marked, the image is fed back to SAM, specifically the SamPredictor initialized with the default model checkpoint, to generate a mask of only the hand or object, which is refined by the annotator input points to hone in on the correct shape. If the hand or object is occluded from the camera, it is marked as concealed. In any case, the selected giver hand, object, and receiver hand masks are heavily validated through a visual review of each color image with the selected mask overlaid onto it. In this validation step, the selected mask is either labeled correct or incorrect by a different annotator than the one who originally selected the mask for that frame. The approximately 15% of masks that were labeled as incorrect were later fixed through the use of an interactive tool. All points clicked and mask indices located at those points are saved in .json files.

**Grasp Taxonomy.** As discussed in Section 4 of the main paper, we categorized all object grasps as per the grasp taxonomy discussed in Cini et al. [3]. Figure 12 shows the breakdown of all 28 grasps with image examples. Power, the use of a power grip, Precision, the use of precision handling, and Intermediate, the use of both power and precision, are the 3 main grasp classifications. Both Power and Precision are broken down into Prismatic and Circular grasp types while Intermediate is broken down into Lateral, Stick, and Writing. Power Prismatic is further broken down into Heavy Wrap, Palmar Wrap, and Medium and Large Wrap. Power Circular has 2 categories, one for disk shaped-objects and the other for sphere-shaped objects. Precision Prismatic, on the other hand, has 4 categories based on the number of fingers used to grasp the object. Prismatic Circular also has an additional Tripod category along with the Disk and Sphere categories. Each lowest-level category has 1 to 4 grasp types that are named with the letter C or F and a number.

# Grasp Type Taxonomy

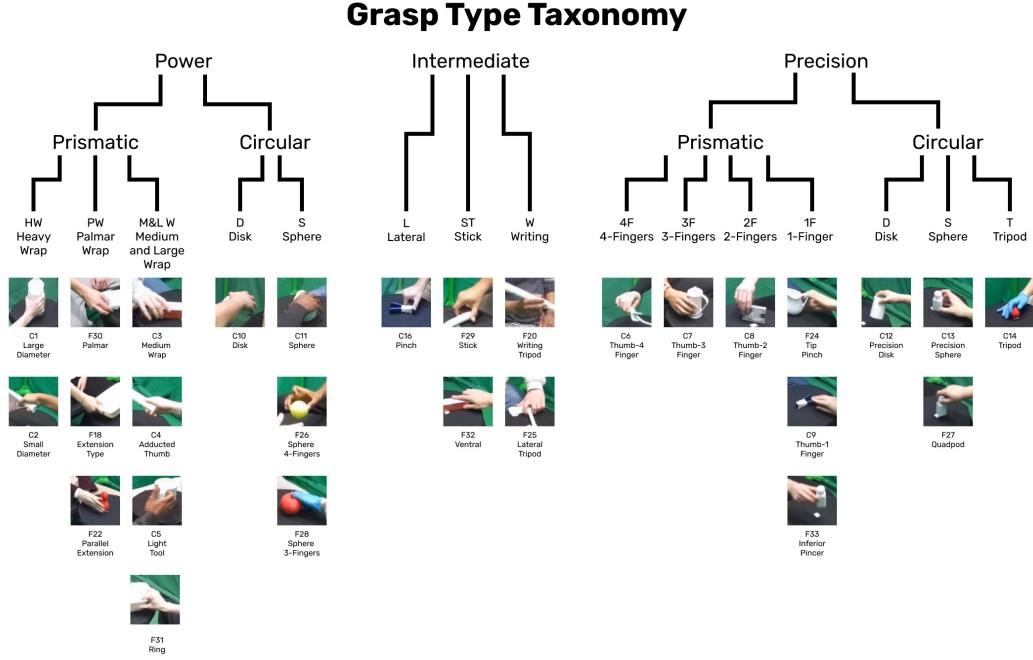

Figure 12: Grasp Type Taxonomy with image examples for all 28 grasp types. The grasp name is under each image.

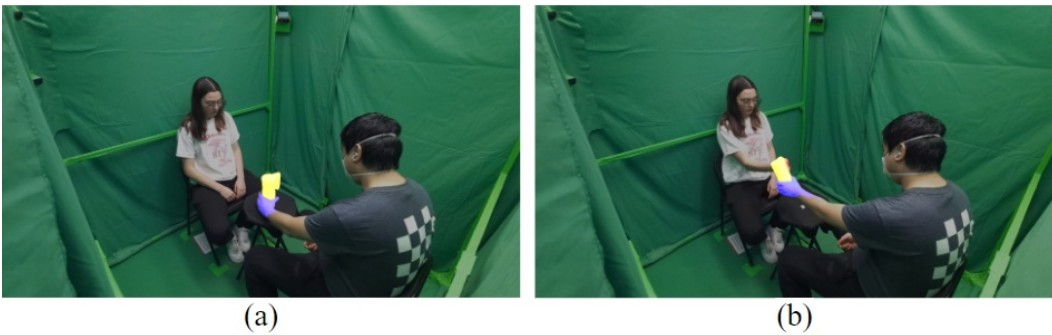

Figure 13: Hand and object mask tracking with the giver's hand in blue, the object in yellow, and the receiver's hand in red. The giver hand mask is increasing up the giver's arm from (a) Frame G, to (b) Frame T.

**Mask Tracking.** The mask tracking approach is robust to enable intermediate frame tracks of hands or objects that may be occluded in a particular key event annotated mask. Track-Anything [10] requires an input mask and a set of images, e.g. for the giver hand track the input would be the Frame G giver hand mask and a list of successive frames from G through R, and for the object track the input would be the Frame G object mask and a list of successive frames from G through R. This approach is also effective in reverse-temporal order, e.g. for the receiver hand track where the input would be the Frame R receiver hand mask and a list of successive frames from R through G. In the event that a giver hand, object, or receiver hand is occluded in a ground truth annotated key event frame, we pass a different key event frame annotation to the tracking program. The likelihood of a particular entity being occluded in all annotated key event frames is low, as the entity and potential sources of occlusion move over time. A potential source for mask tracking error is due to the tendency of a track to slowly expand over successive frames, especially for hand masks when the participant was not wearing long sleeves, as shown in Figure 13. All tracked masks for objects, giver hands, and receiver hands for single interactions are compiled in forward-temporal order, i.e. G to R order, and saved in a compressed .npz file.

**Network and Training Details** For `o2gg` and `g2rg`, the hyperparameters of their base network PoinTr [11] were used as default, i.e., 300 epochs and learn rate of 0.0005, and a batch size of 24 was used as automatically determined by the PoinTr implementation. We used the PoinTr implementation as is. For `g2rt`, the learning rate was kept to the default of Informer [13] at 0.0001, and the batch size was increased to 64. The number of epochs were set to 1,000. Since we addressed spatial trajectory generation, only positional encodings were used for Informer, and temporal embeddings were eliminated. For `o2or`, batch size of 24 and learn rate of 0.001 were used. Due to the smaller network size, the training error declined rapidly, as such training epochs were set to 80. For `o2or` the original PointNet [8] encoder was used to generate a 1,024 global feature vector, which was fed to a (1,024,128,64,4) multi-layer perceptron that yielded the x, y, z, and w components of the rotation quaternion.

# 6   Computing and Training Details

**Computing Resources and Annotation Time Estimates.** Computing resources were used for the following non-experimental purposes. The extraction and synchronization processes were CPU-based, and ran for approximately 240 hours spread across 3 computers with AMD Ryzen 2700X CPUs. The full-scene Segment Anything Model (SAM) [6] segmentation for all Kinect color images ran for approximately 3,000 hours spread across 9 NVIDIA 3090 and 2 NVIDIA 3090Ti GPUs. Intermediate frame entity mask tracking ran for approximately 450 hours spread across 5 NVIDIA 3090 GPUs. Point cloud processing were CPU-intensive, and ran for approximately 1,500 hours across 20 computers with AMD Ryzen 2700X CPUs. Annotators spent approximately 580 hours performing all annotations for the dataset.

**Computing Resources for Experimental Results** We trained and tested the neural networks for `o2gg` and `g2rg` using our own server with four (4) NVIDIA M40 GPUs, two (2) Intel Xeon E5-2640 v4 CPUs, and 128GB of RAM. We trained and tested the neural networks for `o2or` using two of our own servers. The first server had two (2) NVIDIA 3090 GPUs, two (2) Intel Xeon E5-2640 v4 CPUs, and 256GB of RAM. The second server was identical to the first, but had 128GB of RAM. We trained and tested the neural networks for `g2rt` using our own server with one (1) NVIDIA 3090, one (1) Intel i5-10600K CPU, and 64GB of RAM.

# 7   Additional Outputs for Experiments

Figures 14 through 19 show further visual results of running `o2gg`, `g2rg`, and `o2or` on complete and partial data. The figures show examples of results that are close to GT on the left, and plausible outputs further from GT on the right.

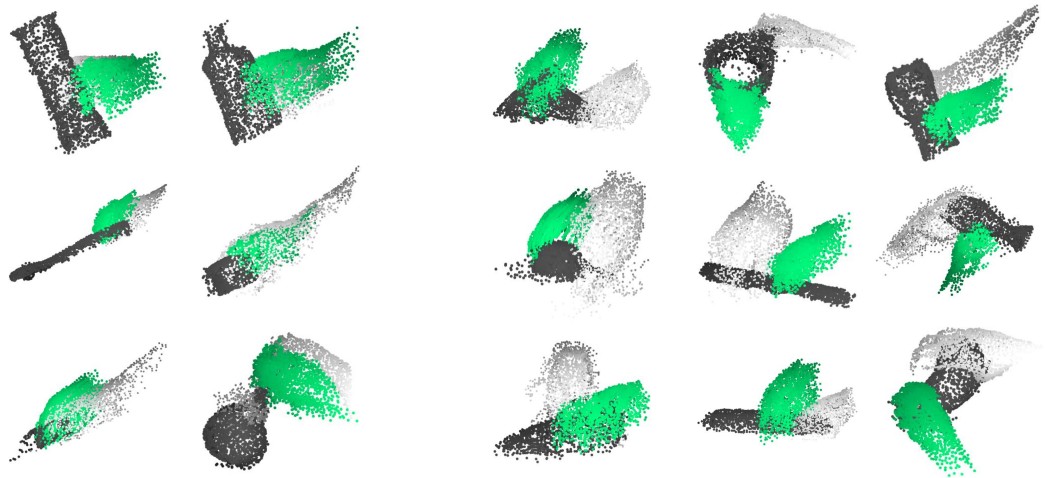

Figure 14: Additional results of o2gg using complete data. Predicted grasp in green versus GT grasp in light gray on input object in dark gray. Examples shown where grasp is close to GT (left) and grasp though deviating from GT is plausible (right).

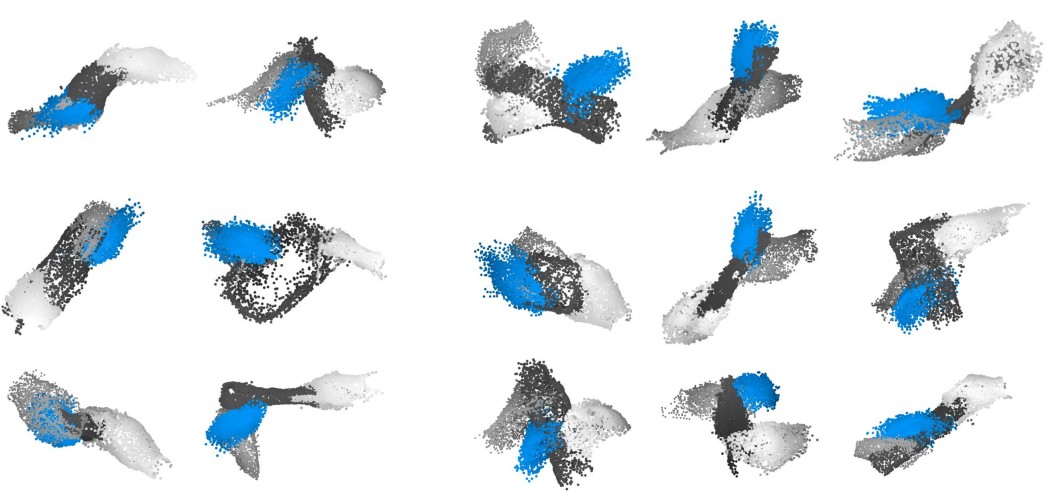

Figure 15: Additional results of g2rg using complete data. Input object and giver are in dark and light gray. Predicted receiver grasp in blue versus GT receiver grasp in medium gray. Examples shown where grasp is close to GT (left) and grasp though deviating from GT is plausible (right).

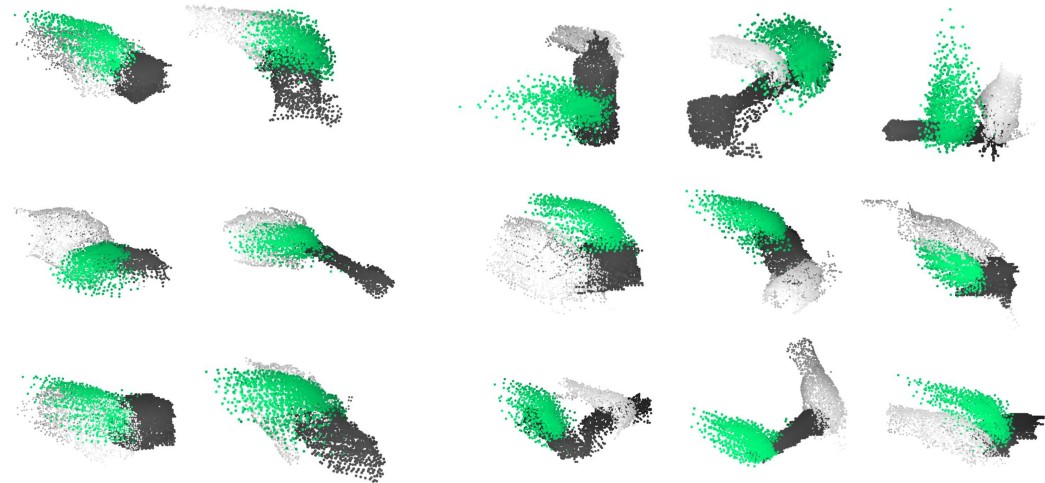

Figure 16: Additional results of o2gg using partial data. Predicted grasp in green versus GT grasp in light gray on input object in dark gray. Examples shown where grasp is close to GT (left) and grasp though deviating from GT is plausible (right).

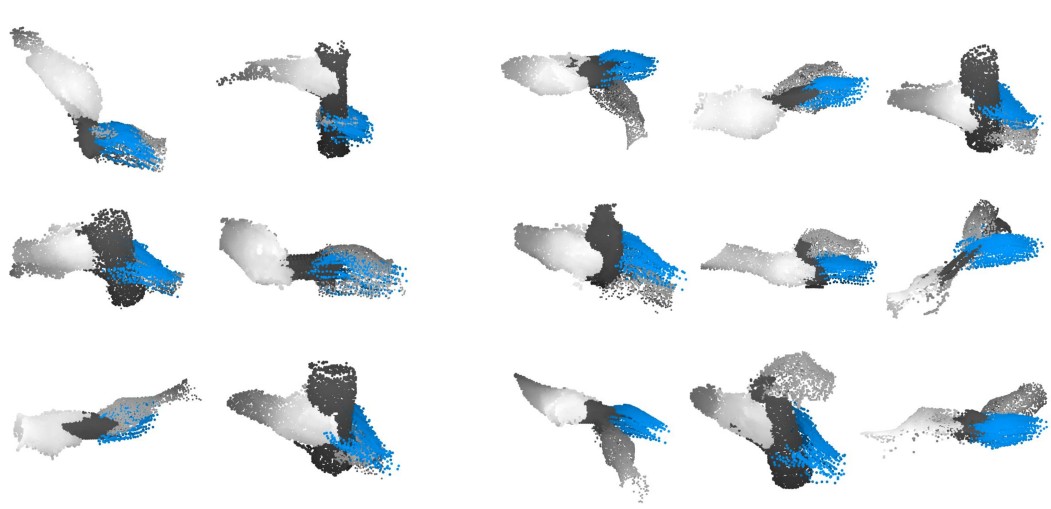

Figure 17: Additional results of g2rg using partial data. Input object and giver are in dark and light gray. Predicted receiver grasp in blue versus GT receiver grasp in medium gray. Examples shown where grasp is close to GT (left) and grasp though deviating from GT is plausible (right).

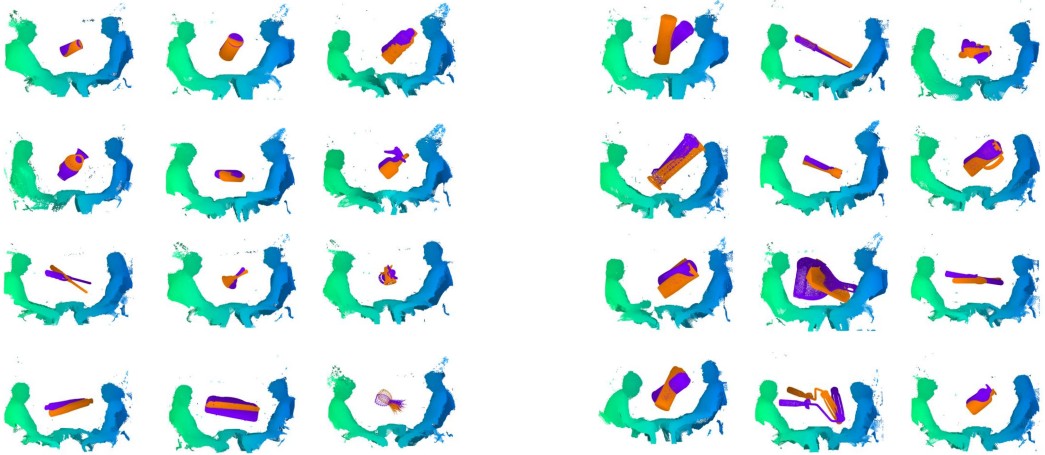

Figure 18: Additional results of o2or using complete data. Predicted orientation in purple versus GT orientation in orange, with examples shown where orientation is close to GT (left) and orientation though deviating from GT is plausible (right).

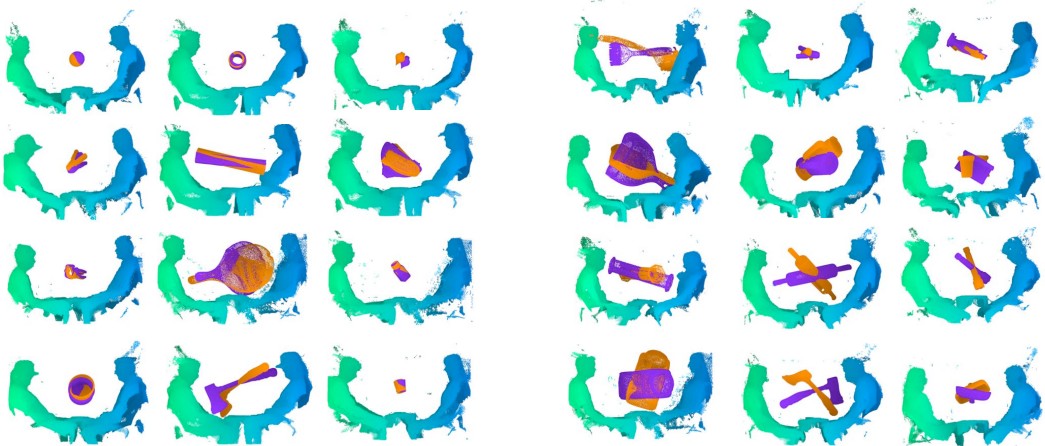

Figure 19: Additional results of o2or using partial data. Predicted orientation in purple versus GT orientation in orange, with examples shown where orientation is close to GT (left) and orientation though deviating from GT is plausible (right).

# 8 Creation of Object Dataset

We collected a set of 136 objects for use in this study. All objects are approximately table-scale, meaning that they could sit unsupported on a table surface. All objects are less than 2.5kg. All objects have at least one dimension that is less than 6 inches to enable unimanual human grasping.

116 objects were store-bought and the remaining 20 were 3D-printed from CAD models of miniatures and perishable items such as fruit. A variety of everyday use items, including 52 of our store-bought objects, have shiny or dark surfaces on which the Kinect's infrared time-of-flight sensor malperforms, yielding poor depth. We coated most of the 52 objects with white matte spray paint, and duct taped a small subset for which spray paint failed to stick.

We use an Einscan-SP 3D scanner with EXScan S to obtain high-fidelity meshes for the 116 store-bought objects as shown in Figure 20. Meshes are bounding-box centered at the origin in their canonical orientation. Meshes are manually cleaned in Autodesk Netfabb to remove any additional parts or scan artifacts that are not present in the real object. Example original and cleaned meshes are shown in Figure 21. The cleaned mesh likely has holes, and is made watertight according to the approach of Stutz and Geiger [9]. We rotated each mesh to a manually determined standard

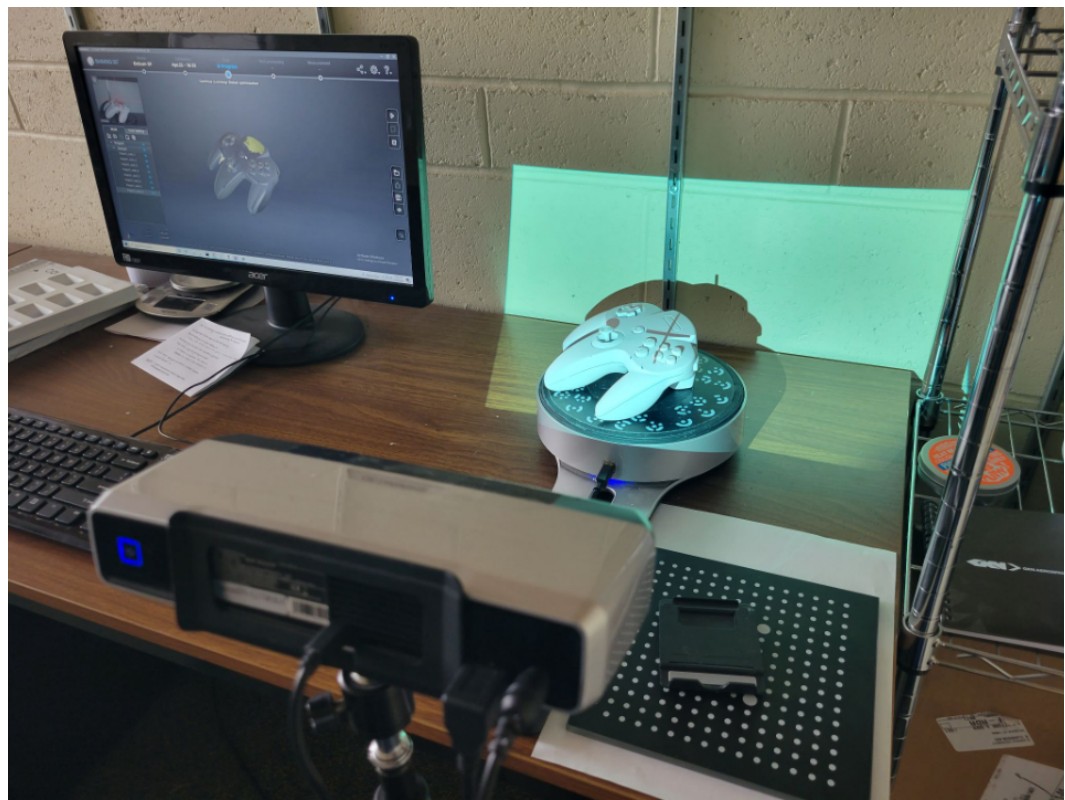

Figure 20: The Einscan-SP 3D scanner (bottom center) with object on turntable being scanned (center right) and EXScan S software compositing the scan (top left).

orientation. For ease of use, the high-fidelity meshes are uniformly simplified down to 40,000 faces using quadratic decimation. We store metadata using the waterproof meshes prior to quadratic decimation. We weigh physical objects on a kitchen scale to obtain mass.

We provide metadata information for all 136 objects used in our work in Table 3. Metadata for objects 116 and 120 is from the original mesh posted on Thingiverse as shown in Table 2. Figures 22 to 38 show renders of the 3D models of the 136 objects, categorized in the 17 aspect-ratio and functionality categories in our work.

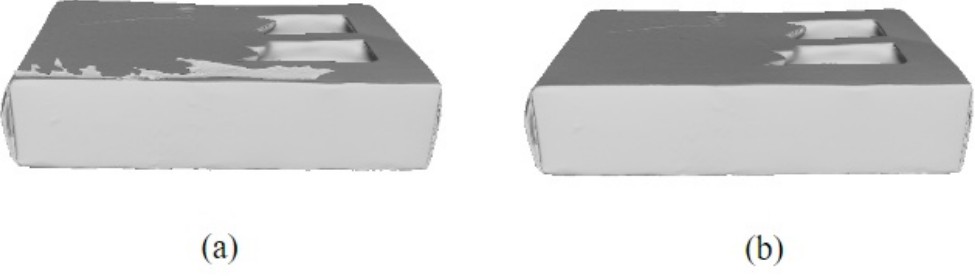

Figure 21: (a) Original mesh from scanner and (b) manually cleaned mesh.

Table 3: Metadata for each object mesh used in HOH.

| ID | Description | Mass (g) | Vertices | Faces | Type | Aspect Ratio |
|---|---|---|---|---|---|---|
| 100 | Rubik's Cube small | 75 | 1,449,255 | 2,506,594 | NFNHV | 1:1-2:1 |
| 102 | Tennis ball | 67 | 1,249,999 | 2,499,994 | NFNHV | 1:1-2:1 |
| 106 | 1" PVC Tee | 67 | 1,235,598 | 2,471,414 | NFNHV | 1:1-2:1 |
| 107 | 1" PVC 90 degree elbow | 53 | 1,249,999 | 2,499,998 | NFNHV | 1:1-2:1 |
| 109 | 1" PVC Coupling | 31 | 1,249,999 | 2,499,998 | NFNHV | 1:1-2:1 |
| 110 | 2" PVC Coupling | 59 | 1,249,995 | 2,500,014 | NFNHV | 1:1-2:1 |
| 114 | Lint roller refill | 80 | 1,241,735 | 2,483,672 | NFNHV | 2:1-3:1 |
| 123 | Joystick | 231 | 1,250,007 | 2,501,696 | NFNHV | 2:1-3:1 |
| T001 | Doll | 165 | 854,375 | 1,709,430 | FNHZ | >3:1 |
| 132 | Lady statue | 176 | 741,212 | 1,482,424 | NFNHV | >3:1 |
| 134 | Power strip tower | 591 | 374,362 | 748,792 | NFNHV | >3:1 |
| 135 | Incense holder | 277 | 1,234,100 | 2,468,686 | NFNHV | >3:1 |
| 200 | Playing Card Deck 4pk | 103 | 1,248,897 | 2,497,828 | NFNHZ | 1:1-2:1 |
| 201 | Lifesaver Candy Box | 242 | 1,228,021 | 2,456,092 | NFNHZ | 1:1-2:1 |
| 202 | Ramekin | 268 | 1,249,999 | 2,499,994 | NFNHZ | 1:1-2:1 |
| 203 | Cookie pan | 973 | 2,034,211 | 4,732,076 | NFNHZ | 1:1-2:1 |
| 204 | Cutting board | 762 | 1,249,999 | 2,499,998 | NFNHZ | 1:1-2:1 |
| 206 | Comp. notebook | 322 | 1,282,894 | 2,715,282 | NFNHZ | 1:1-2:1 |
| 207 | Spiral notebook | 49 | 1,249,818 | 2,502,330 | NFNHZ | 1:1-2:1 |
| 211 | Gift box | 103 | 1,249,995 | 2,500,030 | NFNHZ | 1:1-2:1 |
| 213 | Butter dish | 216 | 1,250,000 | 2,499,996 | NFNHZ | 2:1-3:1 |
| 216 | Microwave omelet cooker | 114 | 1,199,080 | 2,398,176 | NFNHZ | 2:1-3:1 |
| 217 | Qtip box (2pk) | 296 | 1,186,889 | 2,373,786 | NFNHZ | 2:1-3:1 |
| 220 | Travel palette | 230 | 1,043,628 | 2,087,264 | NFNHZ | 2:1-3:1 |
| 222 | iPhone Case | 29 | 1,095,691 | 2,191,526 | NFNHZ | 2:1-3:1 |
| 223 | iPad case | 305 | 1,246,934 | 2,521,020 | NFNHZ | 2:1-3:1 |
| 224 | Level | 52 | 580,234 | 1,160,500 | NFNHZ | >3:1 |
| 225 | Aluminum foil box | 181 | 926,667 | 1,853,330 | NFNHZ | >3:1 |
| 301 | Glass jar/lid | 738 | 1,249,999 | 2,499,994 | FNHV | 1:1-2:1 |
| 303 | 1Gal storage container | 173 | 1,249,990 | 2,500,008 | FNHV | 1:1-2:1 |
| 304 | Ball pencil sharpener | 154 | 249,156 | 498,308 | FNHV | 1:1-2:1 |
| 305 | Tin of frosting | 386 | 1,249,999 | 2,499,998 | FNHV | 1:1-2:1 |
| 308 | Picture frame 11x14 | 764 | 1,250,013 | 2,500,070 | FNHV | 1:1-2:1 |
| 309 | Pringles can (short) (12pk) | 104 | 1,249,964 | 2,500,044 | FNHV | 1:1-2:1 |
| 310 | Campbell's soup 10.75oz (8pk) | 351 | 1,209,073 | 2,418,368 | FNHV | 1:1-2:1 |
| 311 | Wrap bandages 2" 6pk | 19 | 1,249,997 | 2,500,006 | FNHV | 1:1-2:1 |
| 314 | Canned Green Beans | 473 | 1,249,882 | 2,501,408 | FNHV | 2:1-3:1 |
| 315 | Whoppers carton | 382 | 1,249,995 | 2,499,998 | FNHV | 2:1-3:1 |
| 316 | Bottle of glue | 179 | 848,235 | 1,696,478 | FNHV | 2:1-3:1 |
| 317 | Laundry crystals | 513 | 936,835 | 1,873,678 | FNHV | 2:1-3:1 |
| 318 | Powerade | 632 | 1,068,025 | 2,136,090 | FNHV | 2:1-3:1 |
| 320 | Soap pump | 102 | 1,061,839 | 2,153,150 | FNHV | 2:1-3:1 |
| 322 | Liquid hand soap 50oz | 2044 | 1,250,029 | 2,500,164 | FNHV | 2:1-3:1 |
| 323 | Hand wash pump | 415 | 1,249,999 | 2,499,998 | FNHV | 2:1-3:1 |
| 326 | Spray bottle | 54 | 1,095,518 | 2,191,040 | FNHV | >3:1 |
| 328 | Lysol disinfectant | 685 | 922,156 | 1,844,316 | FNHV | >3:1 |
| 329 | Pringles can | 198 | 1,127,367 | 2,254,946 | FNHV | >3:1 |
| 330 | Spray cheese | 293 | 1,106,393 | 2,213,170 | FNHV | >3:1 |
| 332 | Water bottle 20oz | 334 | 1,249,352 | 2,499,096 | FNHV | >3:1 |
| 333 | Water bottle 17oz | 300 | 734,182 | 1,468,372 | FNHV | >3:1 |
| 334 | Tennis ball container | 47 | 1,249,692 | 2,500,012 | FNHV | >3:1 |
| 335 | Salt/pepper shakers | 107 | 1,195,377 | 2,390,772 | FNHV | >3:1 |
| F001 | Macaroni & Cheese Box (3pk) | 369 | 1,249,997 | 2,499,998 | FNHZ | 1:1-2:1 |
| 402 | Cheese dip | 282 | 1,237,936 | 2,476,016 | FNHZ | 1:1-2:1 |
| 403 | Pastry scraper | 204 | 150,009 | 2,300,020 | FNHZ | 1:1-2:1 |
| 404 | Salad hands | 107 | 475,852 | 951,728 | FNHZ | 1:1-2:1 |
| 406 | Tupperware medium | 65 | 1,249,982 | 2,500,098 | FNHZ | 1:1-2:1 |
| 408 | Wireless comp. mouse | 51 | 1,250,000 | 2,499,996 | FNHZ | 1:1-2:1 |
| 410 | Spring clamp | 67 | 1,102,878 | 2,205,778 | FNHZ | 1:1-2:1 |
| 411 | Pringles pack (small) 18pk | 29 | 1,237,226 | 2,474,582 | FNHZ | 1:1-2:1 |
| 412 | 4" paint brush | 83 | 579,790 | 1,161,096 | FNHZ | 2:1-3:1 |
| 415 | 6oz can | 180 | 1,240,341 | 2,480,686 | FNHZ | 2:1-3:1 |

| 416 | Long tissue box | 299 | 1,249,999 | 2,499,998 | FNHZ | 2:1-3:1 |
|------|------------------|------|-----------|-----------|---------|---------|
| 417 | Peeler | 37 | 358,598 | 720,790 | FNHZ | 2:1-3:1 |
| 418 | Pizza cutter | 55 | 793,896 | 1,588,056 | FNHZ | 2:1-3:1 |
| 419 | Garden trowel | 187 | 390,889 | 781,782 | FNHZ | 2:1-3:1 |
| 421 | Cleaver | 399 | 380,195 | 760,131 | FNHZ | 2:1-3:1 |
| 423 | Hatchet | 950 | 636,305 | 1,272,694 | FNHZ | 2:1-3:1 |
| 424 | Stapler | 150 | 967,171 | 1,934,418 | FNHZ | >3:1 |
| 426 | Flat iron | 161 | 481,708 | 963,424 | FNHZ | >3:1 |
| O010 | Big Eraser | 214 | 1,237,999 | 2,476,030 | FNHZ | >3:1 |
| 430 | Hand rake | 228 | 418,414 | 836,844 | FNHZ | >3:1 |
| 432 | Ice pick | 93 | 324,325 | 648,680 | FNHZ | >3:1 |
| C003 | Ice Cube Tray | 103 | 1,241,945 | 2,484,010 | NFNHZ | 2:1-3:1 |
| 434 | Rubber scraper | 39 | 293,714 | 587,428 | FNHZ | >3:1 |
| 435 | Curling iron | 196 | 370,451 | 740,898 | FNHZ | >3:1 |
| 500 | Measuring cup | 79 | 1,249,995 | 2,500,006 | FHV | 1:1-2:1 |
| 502 | Ceramic mug (2pk) | 358 | 1,249,999 | 2,499,998 | FHV | 1:1-2:1 |
| 503 | Clorox 64oz | 2069 | 1,249,994 | 2,500,000 | FHV | 1:1-2:1 |
| 505 | Coffee mug (small, handle) | 233 | 1,249,994 | 2,500,008 | FHV | 1:1-2:1 |
| 507 | Cream holder | 535 | 1,250,016 | 2,500,092 | FHV | 1:1-2:1 |
| 508 | Flour sifter | 139 | 1,229,875 | 2,459,966 | FHV | 1:1-2:1 |
| 509 | Hot glue gun | 86 | 1,008,984 | 2,017,984 | FHV | 1:1-2:1 |
| 511 | Travel mug 7 (handle) | 378 | 1,158,350 | 2,316,908 | FHV | 1:1-2:1 |
| 513 | Saucepan 2 | 617 | 1,118,018 | 2,236,046 | FHV | 2:1-3:1 |
| 514 | Hand bell | 74 | 769,846 | 1,540,082 | FHV | 2:1-3:1 |
| 516 | Spatula/turner | 146 | 316,995 | 610,792 | FHV | 2:1-3:1 |
| 517 | Clorox spray bottle | 739 | 978,053 | 1,956,130 | FHV | 2:1-3:1 |
| 518 | Tide spray bottle | 500 | 1,034,111 | 2,068,234 | FHV | 2:1-3:1 |
| 519 | Glass pitcher 60oz | 614 | 1,249,999 | 2,499,998 | FHV | 2:1-3:1 |
| 520 | Travel mug 4 (handle) | 425 | 1,282,759 | 2,500,002 | FHV | 2:1-3:1 |
| 522 | Travel mug 6 (handle) | 449 | 1,280,815 | 2,499,998 | FHV | 2:1-3:1 |
| 525 | Thin flash light | 55 | 514,793 | 1,029,594 | FHV | >3:1 |
| 526 | Long lighter (6pk) | 35 | 199,958 | 399,916 | FHV | >3:1 |
| 527 | Coffee press | 234 | 1,242,822 | 2,485,672 | FHV | >3:1 |
| 528 | Handheld grater/zester | 93 | 871,379 | 1,846,140 | FHV | >3:1 |
| 530 | Travel mug 3 (handle) | 392 | 1,249,994 | 2,500,000 | FHV | >3:1 |
| 531 | Travel mug 10 (handle) | 374 | 1,249,915 | 2,500,098 | FHV | >3:1 |
| 533 | Toilet brush | 166 | 515,317 | 1,031,586 | FHV | >3:1 |
| 534 | Grill brush | 313 | 743,919 | 1,488,358 | FHV | >3:1 |
| 601 | Pastry cutter | 168 | 1,232,141 | 2,465,350 | FHZ | 1:1-2:1 |
| 602 | Pizza peel | 375 | 993,058 | 1,987,112 | FHZ | 1:1-2:1 |
| 603 | Paint roller frame | 212 | 552,552 | 1,105,252 | FHZ | 1:1-2:1 |
| 605 | Ping pong paddle (2) | 132 | 785,606 | 1,571,540 | FHZ | 1:1-2:1 |
| 606 | Pickleball paddle (2) | 242 | 1,258,015 | 2,525,668 | FHZ | 1:1-2:1 |
| 607 | Locking c-clamp pliers 6" | 51 | 906,725 | 1,813,462 | FHZ | 1:1-2:1 |
| 610 | Dustpan | 120 | 1,246,308 | 2,495,334 | FHZ | 1:1-2:1 |
| 611 | Brush ^ | 180 | 1,240,321 | 2,480,836 | FHZ | 1:1-2:1 |
| 701 | Xbox controller | 206 | 1,250,000 | 2,499,996 | other | 1:1-2:1 |
| 702 | Playstation controller | 229 | 1,235,324 | 2,470,656 | other | 1:1-2:1 |
| 703 | N64 controller | 169 | 1,249,999 | 2,500,018 | other | 1:1-2:1 |
| 704 | Gamecube controller | 156 | 1,248,340 | 2,496,698 | other | 1:1-2:1 |
| 706 | SNES controller (2pk) | 67 | 935,443 | 1,870,902 | other | 1:1-2:1 |
| 708 | Wii classic controller (2pk) | 114 | 1,250,031 | 2,500,106 | other | 1:1-2:1 |
| 713 | Loaf pan | 916 | 1,250,111 | 2,500,444 | other | 1:1-2:1 |
| 714 | Rolling pin | 573 | 761,806 | 1,525,708 | other | 1:1-2:1 |
| 104 | Apple | 88 | 5,644 | 11,285 | NFNHV | 1:1-2:1 |
| 105 | Bell pepper | 87 | 21,094 | 42,184 | NFNHV | 1:1-2:1 |
| 115 | Santa | 41 | 246,876 | 493,744 | NFNHV | 2:1-3:1 |
| 116 | Deco vase | 146 | 103,896 | 207,788 | NFNHV | 2:1-3:1 |
| 118 | Column pot | 54 | 574 | 1,144 | NFNHV | 2:1-3:1 |
| 120 | Baby yoda statue | 56 | 778,292 | 1,556,670 | NFNHV | 2:1-3:1 |
| 121 | Short spiral ornament | 21 | 44,752 | 89,500 | NFNHV | 2:1-3:1 |
| 122 | Candle lantern | 112 | 86,431 | 173,082 | NFNHV | 2:1-3:1 |
| 124 | Pineapple | 74 | 8,108 | 16,278 | NFNHV | >3:1 |
| 126 | Zucchini | 42 | 340,012 | 680,020 | NFNHV | >3:1 |

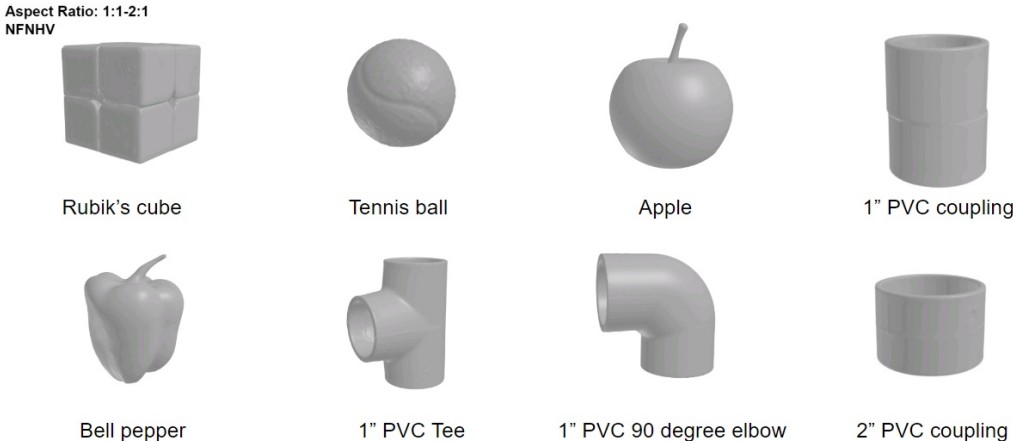

**Aspect Ratio: 1:1-2:1**
**NFNHV**

| | | | |
|---|---|---|---|
| Rubik's cube | Tennis ball | Apple | 1" PVC coupling |
| Bell pepper | 1" PVC Tee | 1" PVC 90 degree elbow | 2" PVC coupling |

Figure 22: Objects from bin 1 which have the aspect ratio 1:1-2:1 and are NFNHV

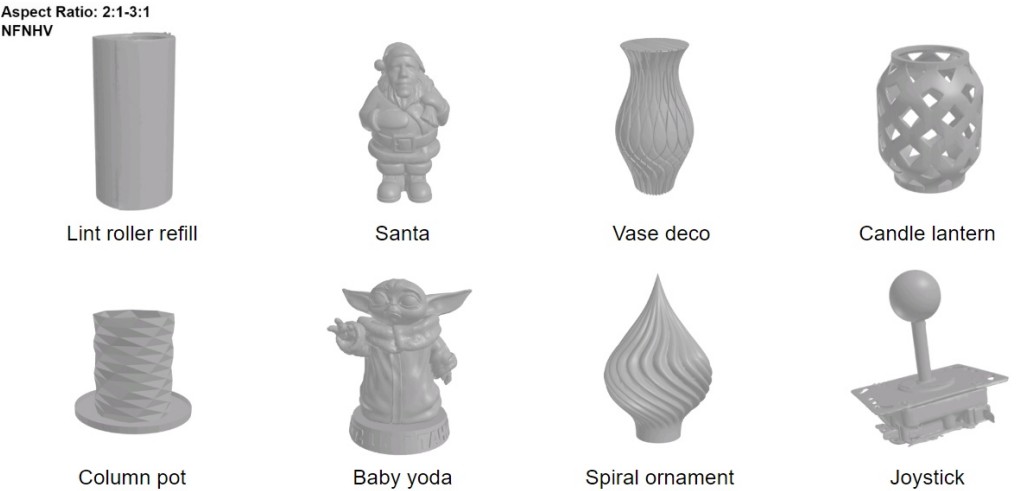

**Aspect Ratio: 2:1-3:1**
**NFNHV**

| | | | |
|---|---|---|---|
| Lint roller refill | Santa | Vase deco | Candle lantern |
| Column pot | Baby yoda | Spiral ornament | Joystick |

Figure 23: Objects from bin 2 which have the aspect ratio 2:1-3:1 and are NFNHV

| | | | | | | |
|---|---|---|---|---|---|---|
| 127 | Twist vase | 35 | 1,077,060 | 2,514,160 | NFNHV | >3:1 |
| 128 | Nefertiti head | 90 | 1,249,999 | 2,499,998 | NFNHV | >3:1 |
| 129 | Statue of liberty | 64 | 101,710 | 203,471 | NFNHV | >3:1 |
| 221 | Balancing bird | 26 | 317,622 | 635,480 | NFNHZ | 2:1-3:1 |
| 228 | Banana | 53 | 5,625 | 11,246 | NFNHZ | >3:1 |
| 232 | Eggplant | 192 | 76,634 | 153,260 | NFNHZ | >3:1 |
| 233 | Saw blade handle | 19 | 2,505 | 5,014 | NFNHZ | >3:1 |
| 234 | Model A Roadster | 51 | 58,191 | 118,146 | NFNHZ | >3:1 |
| 235 | Star wars ship model | 42 | 95,825 | 191,704 | NFNHZ | >3:1 |
| 236 | SUV model | 37 | 26,012 | 52,036 | NFNHZ | >3:1 |

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

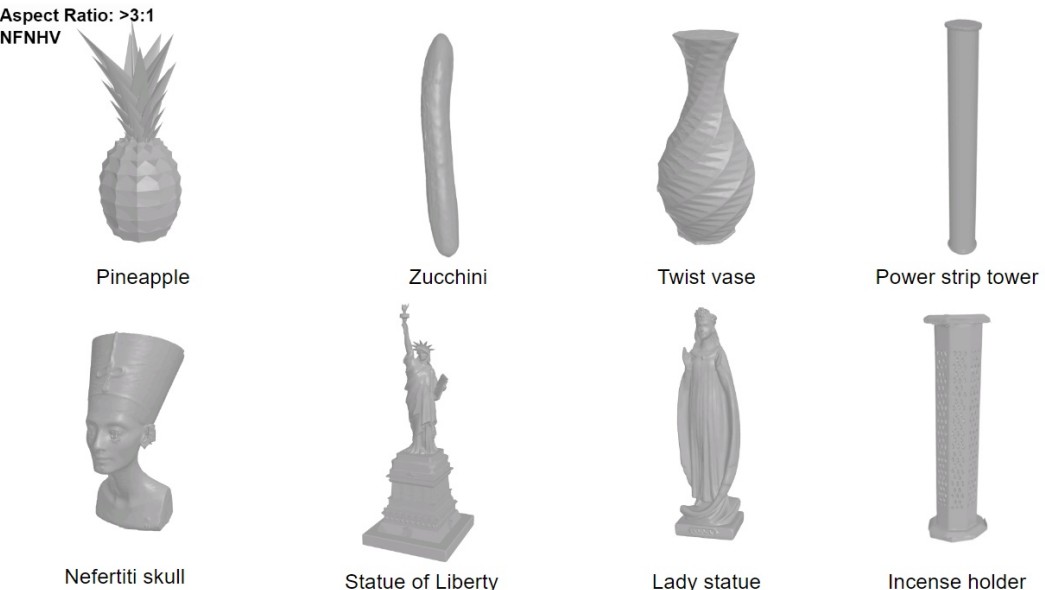

Figure 24: Objects from bin 3 which have the aspect ratio >3:1 and are NFNHV

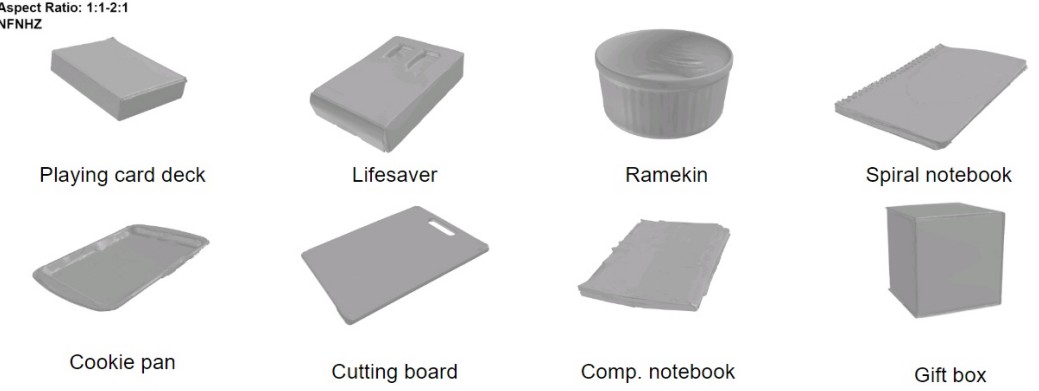

Figure 25: Objects from bin 4 which have the aspect ratio 1:1-2:1 and are NFNHZ

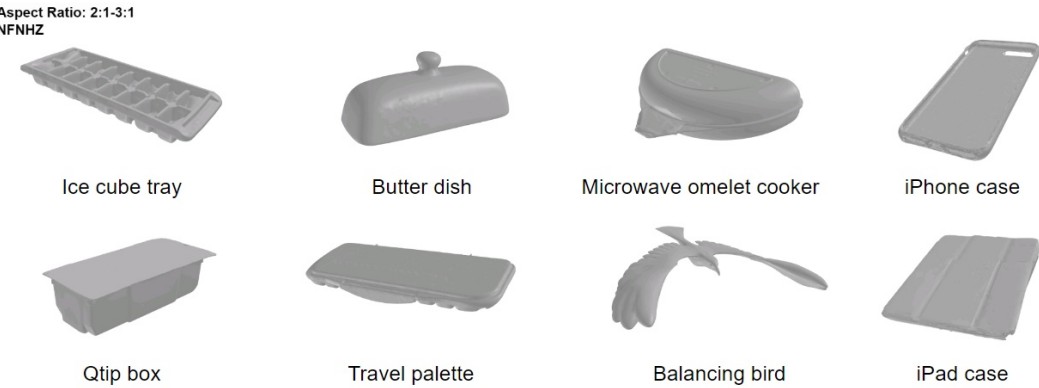

Figure 26: Objects from bin 5 which have the aspect ratio 2:1-3:1 and are NFNHZ

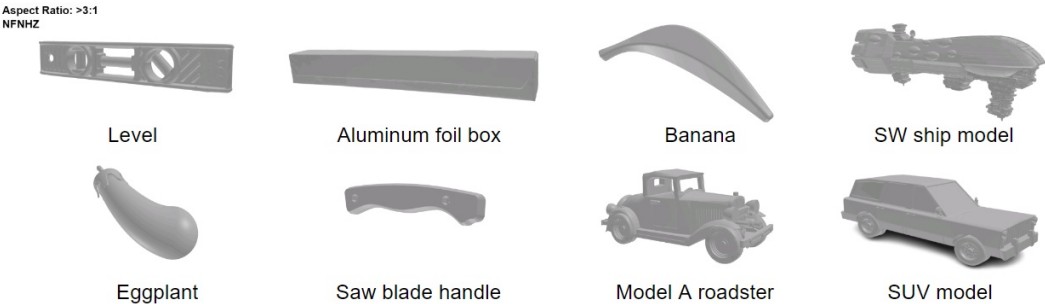

Figure 27: Objects from bin 6 which have the aspect ratio >3:1 and are NFNHZ

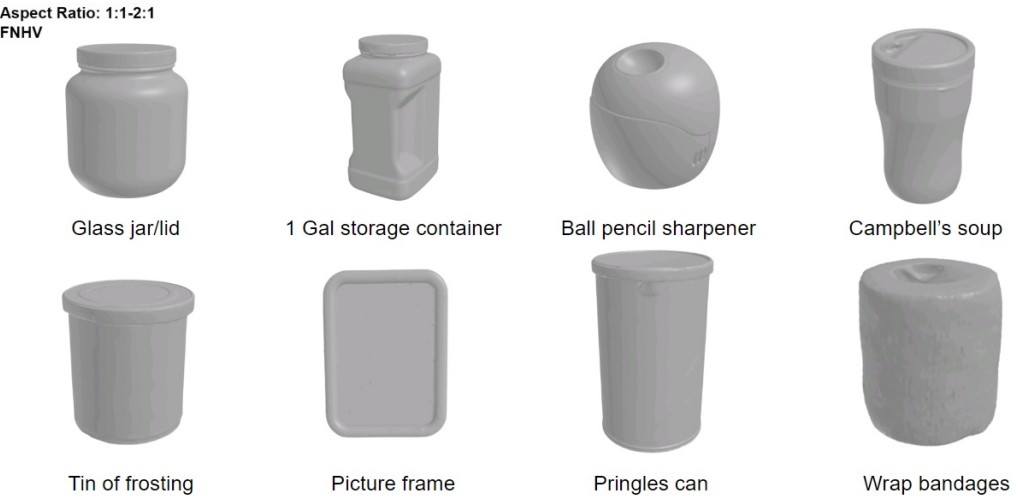

Figure 28: Objects from bin 7 which have the aspect ratio 1:1-2:1 and are FNHV

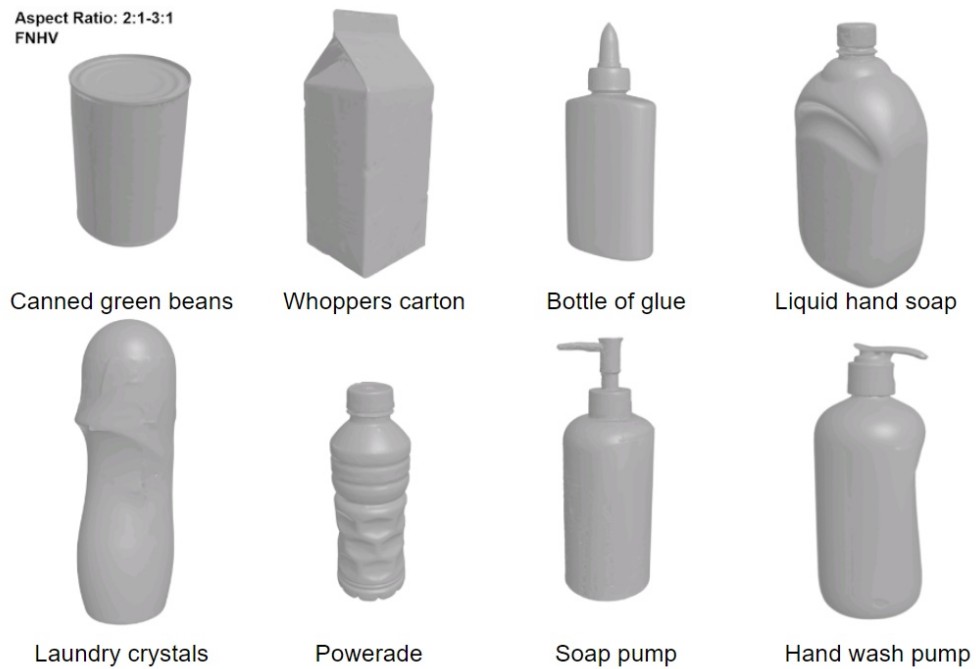

Figure 29: Objects from bin 8 which have the aspect ratio 2:1-3:1 and are FNHV

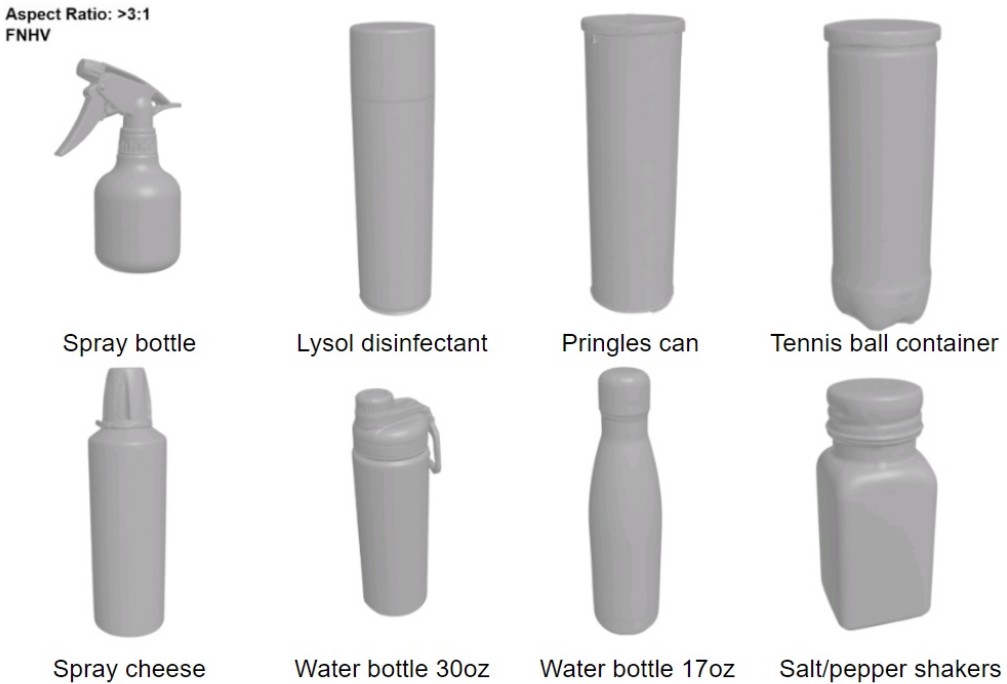

Figure 30: Objects from bin 9 which have the aspect ratio >3:1 and are FNHV

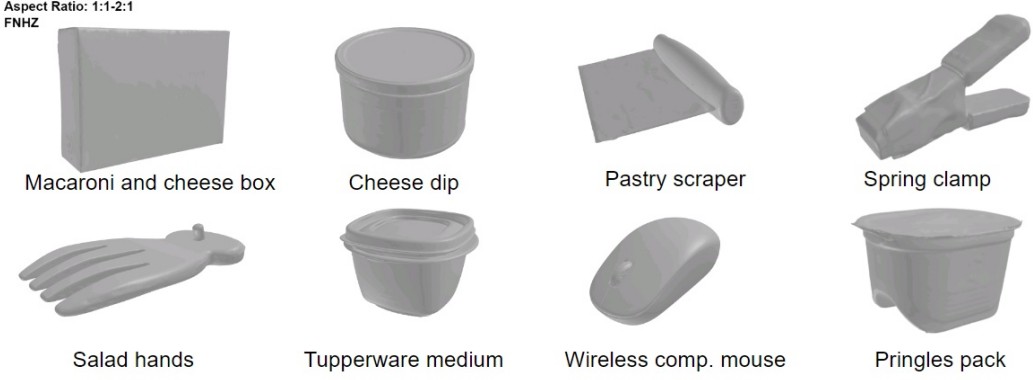

Figure 31: Objects from bin 10 which have the aspect ratio 1:1-2:1 and are FNHZ

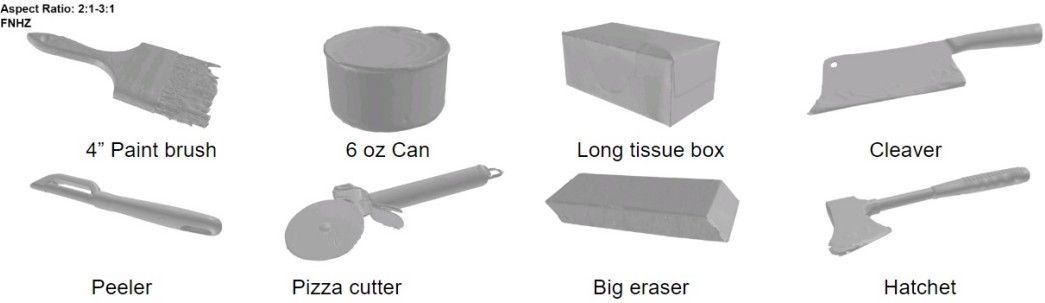

Figure 32: Objects from bin 11 which have the aspect ratio 2:1-3:1 and are FNHZ

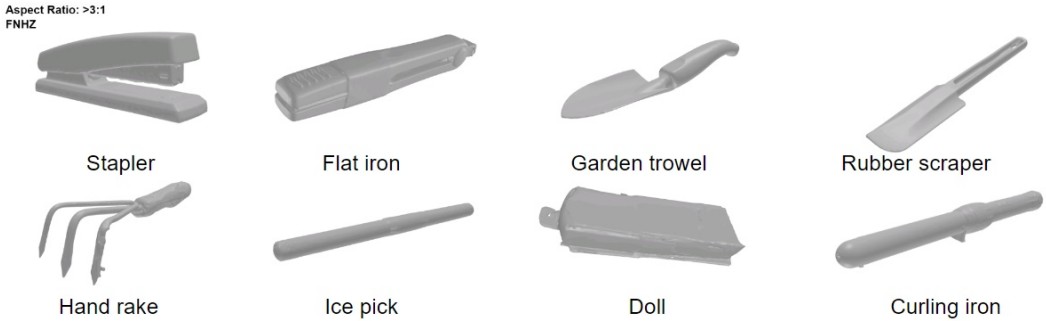

Figure 33: Objects from bin 12 which have the aspect ratio >3:1 and are FNHZ

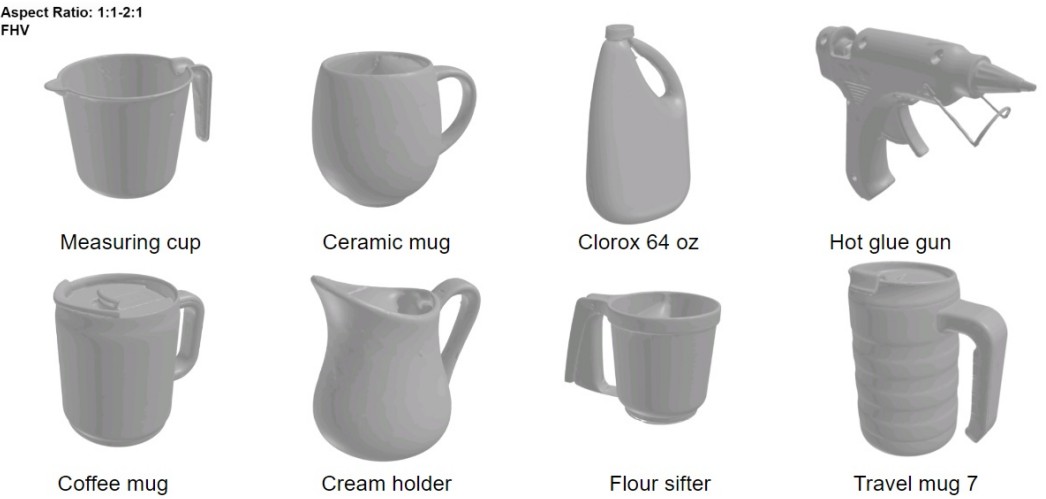

Figure 34: Objects from bin 13 which have the aspect ratio 1:1-2:1 and are FHV

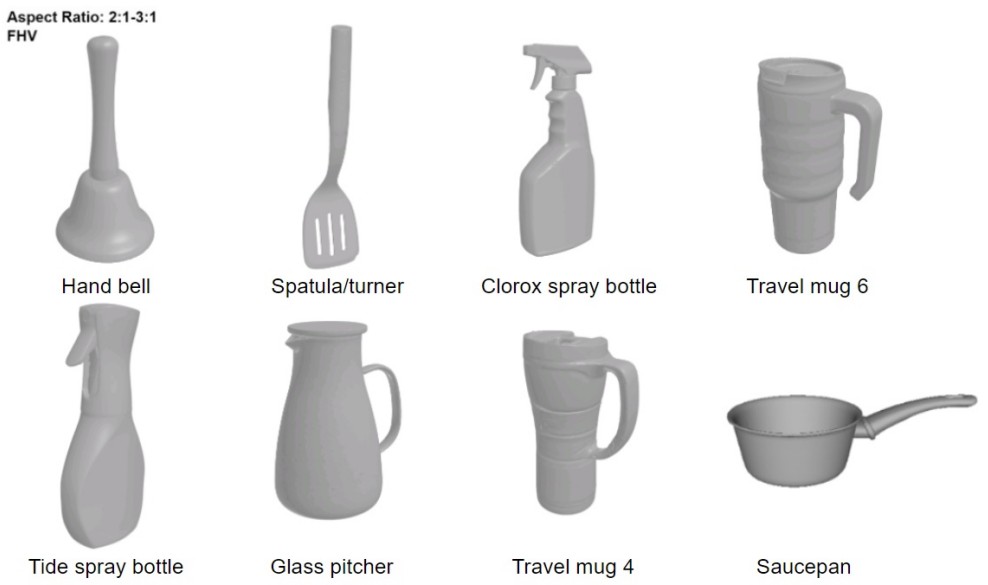

Figure 35: Objects from bin 14 which have the aspect ratio 2:1-3:1 and are FHV

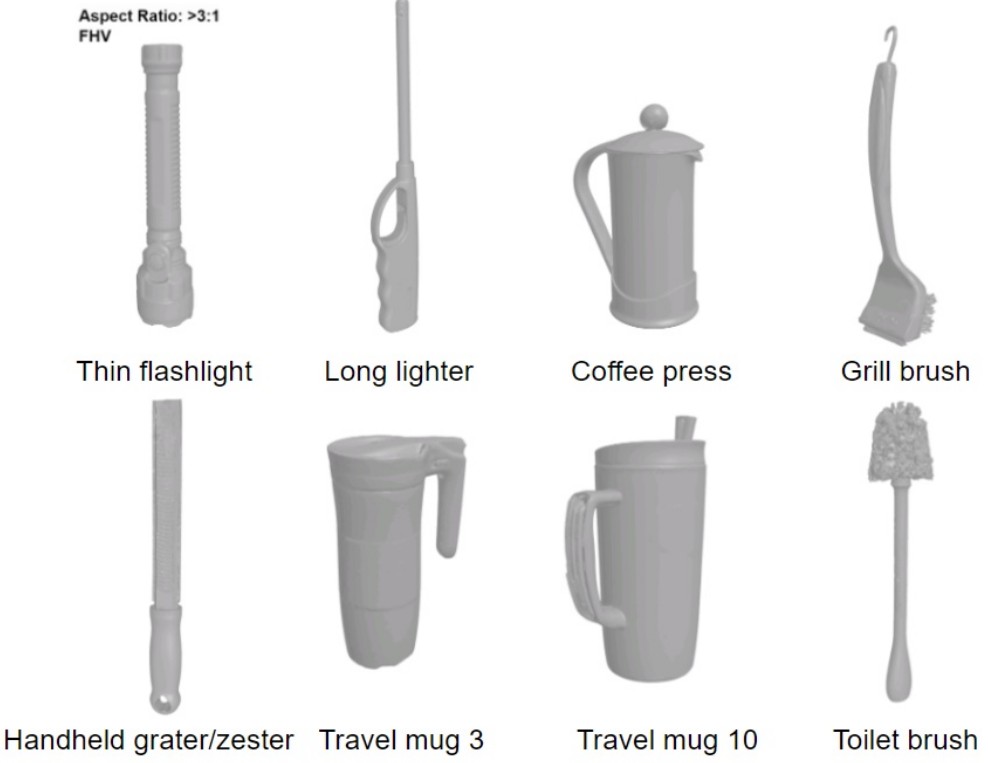

Figure 36: Objects from bin 15 which have the aspect ratio >3:1 and are FHV

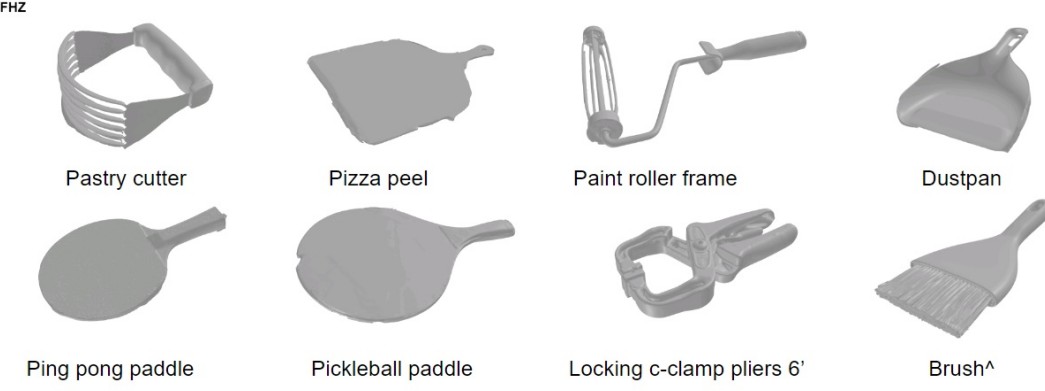

Figure 37: Objects from bin 16 which are FHZ

weird, two handle, etc

| | | | |
|---|---|---|---|
| Xbox controller | PlayStation controller | N64 controller | Loaf pan |
| GameCube controller | SNES controller | Wii classic controller | Rolling pin |

Figure 38: Objects from bin 17 which are classified as "other"

[3] F Cini, V Ortenzi, P Corke, and MJSR Controzzi. On the choice of grasp type and location when handing over an object. *Science Robotics*, 4(27), 2019.

[4] M. Musy et al. vedo, a python module for scientific analysis and visualization of 3D objects and point clouds.

[5] Timnit Gebru, Jamie Morgenstern, Briana Vecchione, Jennifer Wortman Vaughan, Hanna M. Wallach, Hal Daumé III, and Kate Crawford. Datasheets for datasets. *CoRR*, abs/1803.09010, 2018.

[6] Alexander Kirillov, Eric Mintun, Nikhila Ravi, Hanzi Mao, Chloe Rolland, Laura Gustafson, Tete Xiao, Spencer Whitehead, Alexander C. Berg, Wan-Yen Lo, Piotr Dollár, and Ross Girshick. Segment anything. *arXiv:2304.02643*, 2023.

[7] MATLAB. *version 7.10.0 (R2010a)*. The MathWorks Inc., Natick, Massachusetts, 2010.

[8] Charles R Qi, Hao Su, Kaichun Mo, and Leonidas J Guibas. Pointnet: Deep learning on point sets for 3d classification and segmentation. *arXiv preprint arXiv:1612.00593*, 2016.

[9] David Stutz and Andreas Geiger. Learning 3d shape completion from laser scan data with weak supervision. In *CVPR*, pages 1955–1964, NJ, 2018. IEEE.

[10] Jinyu Yang, Mingqi Gao, Zhe Li, Shang Gao, Fangjing Wang, and Feng Zheng. Track anything: Segment anything meets videos, 2023.

[11] Xumin Yu, Yongming Rao, Ziyi Wang, Zuyan Liu, Jiwen Lu, and Jie Zhou. Pointr: Diverse point cloud completion with geometry-aware transformers. In *CVPR*, pages 12498–12507. IEEE, 2021.

[12] Z. Zhang. A flexible new technique for camera calibration. *IEEE Transactions on Pattern Analysis and Machine Intelligence*, 22(11):1330–1334, 2000.

[13] Haoyi Zhou, Shanghang Zhang, Jieqi Peng, Shuai Zhang, Jianxin Li, Hui Xiong, and Wancai Zhang. Informer: Beyond efficient transformer for long sequence time-series forecasting. In *Proceedings of the AAAI conference on artificial intelligence*, volume 35, pages 11106–11115, 2021.