# OpenReview forum: "HOH: Markerless Multimodal Human-Object-Human Handover Dataset with Large Object Count"
_NeurIPS.cc/2023/Track/Datasets_and_Benchmarks — NeurIPS 2023 Datasets and Benchmarks Poster_

### Official Review · Reviewer_e1Fp · 2023-07-05

**Rating:** 7
**Confidence:** 4
**Correctness:** What if OpenPose or SAM produce wrong…
**Clarity:** Yes

**Strengths:**

This dataset introduces a much larger scale of assets for handover studies. Also, the dataset is captured in a markerless environment, which makes appearances of images look more natural than marker-based ones.

**Additional Feedback:**

Please see the weaknesses.

**Documentation:**

Yes

**Limitations:**

Yes

**Opportunities For Improvement:**

1. Although Table 1 compares the proposed HOH dataset to existing ones, it simply compares the quantity of assets. More in-depth comparison, such as visualized t-sne of datasets, should be shown to demonstrate that the proposed HOH dataset includes *more diverse* assets (not simply many assets).

2. Where can this dataset be used is not experimentally shown. All descriptions of this paper are about 1) how to capture this dataset and 2) statistics of this dataset. The authors should have shown some systems trained on this dataset performs better on held-out evaluation benchmarks compared to not using this dataset.

3. Not sure about the benefit of the markerless capture. The authors argued that one of the main advantages of this dataset is a markerless capture. Definitely, without markers and specially designed suits, images look more natural. But as Figure 2 shows, all images are captured with a green background and from several fixed viewpoints. I do not think vision-based systems, trained on this dataset, can generalize well to in-the-wild images due to the limited diversity of image appearances of this dataset. Then, what is the benefit of the markerless capture? This also should be justified by experiments, for example, training a system on existing mocap data and the proposed data and test it on held-out benchmarks.

4. Lack of 3D poses of hands and objects. As the authors mentioned in Section 5, the proposed dataset do not have 3D poses of hands and objects. For the handover analysis, both 3D poses of hands and objects are critical, which makes this dataset less attractive. I guess using marker-based captures, getting 3D poses of hands and objects are relatively easier than the markerless capture, which again makes me confused about the benefit of the markerless capture.

**Relation To Prior Work:**

No. Please see the first weakness.

**Summary And Contributions:**

This paper introduces a new dataset, HOH, which includes a large scale of handover data. The dataset includes multi-view RGBD, skeleton, point cloud, grasp type, handedness, 2D/3D segmentation masks, and 3D object models. The main strong points of this dataset compared to existing ones are markerless capture and large scale.

---

> ### Author Response · Authors · 2023-08-22
> **Response to e1Fp**
>
> **t-SNE:**
> We have generated t-SNE maps for the 3D models of all the objects used in our dataset and shown distributions of the top 3 coefficients, labeled by form/function category in the paper in Fig. 7. We also have generated t-SNE maps for the giver, object, and receiver trajectories for all interactions. For comparison, we show t-SNE maps for giver and receiver hand mocap trajectories from the Carfi dataset, showing a greater spread over space for our dataset compared to Carfi. To ensure normalization across both datasets, we mean-center, scale-normalize, and Procrustes-align the trajectories within the dataset, since Carfi dataset users are displaced in space (and rigid transformations are easy to emulate for us synthetically).
>
> **3D pose alignment**
> At this time, we have conducted 6DOF 3D pose alignment of the object’s 3D model to all frames from G to R for all handover interactions in the dataset, except one interaction where the hand obscures a significant portion of the object throughout. First, we conduct a frame-to-frame iterative closest point (ICP) alignment of the object point cloud to estimate inter-frame cloud-to-cloud transformations of the object point cloud for all frames, linking all frames from G to R. Next, we conduct an automated alignment of the 3D model to the object point cloud in the G frame. We sample a diverse range of orientations to ensure exhaustive SO(3) coverage. We transform each 3D model using each orientation and fine-tune the transformed model's alignment to the object point cloud using ICP. We choose the 3D model alignment with the smallest ICP distance. We use the inter-frame transformations to align the 3D model to all frames from G to R. Example frames from the alignment are shown in the paper in Fig. 4. We also show example videos of the alignment at the shared webpage. We have now updated various regions of the paper, including the list of contributions in Sec 1 to discuss the 6DOF pose. Due to our small team size, we do not have the capacity to align hand poses at this time, however, the object poses enable us to create neural networks for object pose estimation at transfer (O2Pose).
>
> **Correctness of SAM and OpenPose:**
> We have manually validated and corrected all SAM masks. We have manually examined all OpenPose video outputs. 95.4% of the OpenPose skeletons are well-aligned, and for only 4.6% the limb is either missing or grossly misaligned. All OpenPose outputs have confidence values available.
>
> **Benefit of 3D markerless motion capture:**
> Our dataset’s primary intended use case is in robotics, particularly in using human-human handover interactions to inform human-robot interactions (HRI). 3D markerless capture for HRI helps develop algorithms that inform robotic manipulators connected with RGB-D sensors on estimating where to grasp for safe human-robot handover. Robots routinely come equipped with RGB-D sensors mounted on end-effectors, platforms, or in the environment. Since RGB-D sensors by design allow point cloud generation, our data enables robotic algorithms that use 3D point clouds of objects to infer properties of handover. Robot givers can learn from the behavior of human givers where to preferentially hold objects, how to move during giving, and how to orient the object at the transfer point when handing the object to a receiver. Robot receivers can use the behavior of human receivers to learn where to grasp objects handed to them by a giver, and what trajectory to navigate in space to remain safe. Lack of background/viewpoint variation does not pose a concern, as 3D viewpoints can be synthetically generated from HOH point clouds to emulate the 3D data from RGB-D sensors. Intended use cases in Section 1 have been expanded (highlighted) to include the above discussion.
>
> **Benchmark evaluations**
> Comments provided in section for overall comments to all reviewers.

---

> > ### Comment · Reviewer_e1Fp · 2023-08-23
> >
> > Dear authors,
> >
> > thanks for the clarifications.
> > I think Figure 7 (b) is very hard to see as the size of points in legend are too small and there are so many colored points in the plot. I think the figure should be modified to make it easier to see.
> > My remaining major concern is about the use case and its experimental demonstration, which seems the authors are going to provide. Are the authors going to provide the benchmark results during the discussion period?

---

> > > ### Author Response · Authors · 2023-08-24
> > > **Response on Figure 7(b) and experimental results**
> > >
> > > Thank you for noting the issue with Figure 7(b). We have now uploaded a revised copy of the paper with the figure improved, and moved to 7(c).
> > >
> > > About the experimental results: That is correct, we plan to generate the proposed results before the discussion phase ends. We have already generated results for O2GGrasp and G2RGrasp for complete objects in the latest revised version of the paper and initiated a new section titled "Experimental Results" (Section 5) in the same. The new Figure 7(b) in the latest version of the paper shows qualitative examples of the O2GGrasp and G2RGrasp output. We currently include quantitative metrics for O2GGrasp and G2RGrasp in the text, however we are working on revising the text to have metrics in a tabular form. We have just completed running the orientation (O2Orientation) prediction network, and are in the process of examining them and generating quantitative results. On compiling them together, we plan to submit a revised update with qualitative and quantitative results later today. Our  grasp and orientation networks from partial data are running on a separate set of GPUs (M40s for grasp from partial and 3090 for orientation from partial), one GPU per network. We expect results on these no later than tomorrow (typically they take much less time than that to train, and we have multiple 3090 and M40 GPUs to run ablation testing in parallel). We are working on setting up our trajectory network today on another parallel couple of 3090 GPUs. Based on our results, we will submit regularly updated versions of our paper over the next several days.

---

> > > > ### Comment · Reviewer_e1Fp · 2023-08-27
> > > >
> > > > Thanks for the additional experiments. I think all of my concerns are addressed. Let me raise my rating.

---

### Official Review · Reviewer_oQbw · 2023-07-06
**Review for HOH 789**

**Rating:** 6
**Confidence:** 3
**Correctness:** Yes
**Clarity:** Yes

**Strengths:**

HOH contains multi-view RGB and depth data, skeletons, fused point clouds,  grasp type and handedness labels, object, giver hand, and receiver hand 2D and 3D segmentations and 3D models for 2,720 handover interactions. As the author declared, HOH is the largest handover dataset in number of objects, participants, pairs with role reversal accounted for, and total interactions captured.

**Additional Feedback:**

None

**Documentation:**

Yes

**Limitations:**

Yes

**Opportunities For Improvement:**

It would be even better if there are some benchmarks in the actual direction to demonstrate what new challenges and opportunities this dataset can provide for future research.

**Relation To Prior Work:**

Yes

**Summary And Contributions:**

In this paper, a markerless 3D multimodal dataset on human-human handovers with136 objects and 20 participant pairs is presented, which contains multi-view RGB and depth data, skeletons, fused point clouds,  grasp type and handedness labels, object, giver hand, and receiver hand 2D and 3D segmentations and 3D models for 2,720 handover interactions.

---

> ### Author Response · Authors · 2023-08-22
> **Benchmark evaluations**
>
> Unfortunately, *there exist no off-the-shelf machine learning algorithms* that use human-human handover data for handover parameter estimation for HRI using 3D data such as from RGB-D sensors (or even 2D image data which may have been adapted for 3D). Only one paper, H2O by Ye et al., 2021, contains an image-based prediction of a handover parameter. Their work is limited to predicting receiver grasp. Neither their dataset nor their algorithm are public. This prevents benchmark evaluations.
>
> Instead, we will provide a novel set of learning-driven algorithms in the revised version of the paper to serve as experimental evaluation  and provide a starting point of experimental results for future research. They also show the benefit of the dataset for HRI. Algorithms:-
>
> 1. Predict human giver hand grasp from object point cloud (O2GGrasp): Helps robotic giver grasp algorithms be biased to grasp near the human giver’s hand. Enables robotic receiver grasp algorithms to plan their grasp to be biased in regions away from the human giver’s hand.
> 2. Predict giver hand motion trajectory from object point cloud (O2GTraj): Helps bias the motion of a robotic giver’s end effector to follow the path of typical human givers rather than plan unsafe trajectories, an issue of ongoing research concern in areas such as reinforcement learning for trajectory planning. Helps robotic receivers plan proactive motion.
> 3. Predict object transfer pose from object point cloud (O2Pose): Helps a robotic giver to orient objects in a human-receiver-preferred manner. Helps a robotic receiver plan to meet the object smoothly for seamless transfer.
> 4. Predict receiver hand grasp from point cloud of giver holding object (G2RGrasp): 2nd method to inform a robot receiver of where to bias grasping during object receipt.
> 5. Predict receiver hand motion trajectory from point cloud of giver holding object (G2RTraj): Helps inform robotic receivers to explore near human trajectories during motion planning, preventing unsafe configurations that could e.g. injure human givers.
>
> Our work adapts point cloud generation neural networks to synthesize giver and receiver output hand point clouds for O2GGrasp/G2RGrasp, and point cloud encoding networks for pose (O2Pose) and PCA coefficients of trajectories in O2GTraj/G2RTraj. We will get results for complete point clouds and partial point clouds for multiple synthetically generated viewpoints emulating data from RGB-D sensors. Our train/test split is 3 objects per form/function bin (51 total) common to train/test, 2 per bin (34) in train only and 3 per bin (51) in test only. We have generated results for O2GGrasp and G2RGrasp using complete point clouds with PoinTr. Metrics are summarized in the paper Sec 5. Sample GT and predicted grasps are shown in Fig. 7. We have resources to conduct proposed experiments by Aug 29. We will reupload the paper with metrics and results. We will also show results of robot grasp planned being guided by neural network outputs to show future research opportunities.

---

> ### Comment · Reviewer_oQbw · 2023-08-24
>
> The authors have addressed some of my concerns. Like another reviewer, my remaining major concern is also about the use case and its experimental demonstration. Are the authors going to provide the benchmark results during the discussion period?

---

> > ### Author Response · Authors · 2023-08-24
> > **Response regarding experimental results**
> >
> > That is correct, we plan to generate the proposed results before the discussion phase ends. We have already generated results for O2GGrasp and G2RGrasp for complete objects in the latest revised version of the paper and initiated a new section titled "Experimental Results" (Section 5) in the same. The new Figure 7(b) in the latest version of the paper shows qualitative examples of the O2GGrasp and G2RGrasp output. We currently include quantitative metrics for O2GGrasp and G2RGrasp in the text, however we are working on revising the text to have metrics in a tabular form. We have just completed running the orientation (O2Orientation) prediction network, and are in the process of examining them and generating quantitative results. On compiling them together, we plan to submit a revised update with qualitative and quantitative results later today. Our  grasp and orientation networks from partial data are running on a separate set of GPUs (M40s for grasp from partial and 3090 for orientation from partial), one GPU per network. We expect results on these no later than tomorrow (typically they take much less time than that to train, and we have multiple 3090 and M40 GPUs to run ablation testing in parallel). We are working on setting up our trajectory network today on another parallel couple of 3090 GPUs. Based on our results, we will submit regularly updated versions of our paper over the next several days.

---

> > > ### Comment · Reviewer_oQbw · 2023-08-27
> > >
> > > Thank you for providing the experimental demonstrations. Most concerns have been addressed. I will increase the rating.

---

### Official Review · Reviewer_MkyD · 2023-07-22
**Large human-object-human (HOH) dataset for detailed analysis**

**Rating:** 8
**Confidence:** 3
**Correctness:** The claims in the paper are correct.
**Clarity:** This paper is well written.

**Strengths:**

The main strength is the large size in objects and human participants, which can contribute to more generalizable  algorithm. Utilization of point clouds instead of marker-based motion capture data can also enable finer analysis and application of identify the motions and interactions of the human, which may be used in more applications.

**Additional Feedback:**

Color image would be very beneficial.

**Documentation:**

The documentation is sufficient for reader to understand and use it.

**Ethics:**

No ethics concerns.

**Limitations:**

The authors have addressed the limitations and societal impacts well.

**Opportunities For Improvement:**

The human subjects can be in different pose than just sitting still. Different poses can create different occlusion and viewpoints. the variety in different poses can force algorithm to be  more robust.

**Relation To Prior Work:**

The difference and relation to prior work is clearly discussed.

**Summary And Contributions:**

This paper describes a dataset consist of human-object-human interaction dataset recorded with multiple depth cameras and generated point clouds. The main contribution of the dataset is the large size in number of participant and number of different objects passed over from one to another, also the use of depth camera for dense point clouds instead of marker based tracking systems.

---

> ### Author Response · Authors · 2023-08-22
> **Different poses**
>
> We appreciate MkyD noting that variety in poses can benefit strengthening algorithms with pose diversity. Unfortunately given the long setup time, the conditions of our IRB protocol, and the time taken to conduct capture and data processing, we lack the bandwidth to capture new data. However, the present dataset can still inform motion for standing or kneeling, by transferring movement from the giver hand, receiver hand, or upper body pose from OpenPose skeletons triangulated to 3D. By reconstructing in 3D, occlusions from diverse viewpoints can be modeled by inserting synthetic cameras.

---

### Author Response · Authors · 2023-08-22
**General Response to All Reviewers**

We thank reviewers for noting our strengths in benefit of markerless data for fine-grained analysis (MkyD), size of our dataset (oQbw and e1Fp), and naturalness of our data (e1Fp).

**Benefit of 3D markerless motion capture (e1Fp):**
Our dataset’s primary intended use case is in robotics, particularly in using human-human handover interactions to inform human-robot interactions (HRI). 3D markerless capture for HRI helps develop algorithms that inform robotic manipulators connected with RGB-D sensors on estimating where to grasp for safe human-robot handover. Robots routinely come equipped with RGB-D sensors mounted on end-effectors, platforms, or in the environment. Since RGB-D sensors by design allow point cloud generation, our data enables robotic algorithms that use 3D point clouds of objects to infer properties of handover. Robot givers can learn from the behavior of human givers where to preferentially hold objects, how to move during giving, and how to orient the object at the transfer point when handing the object to a receiver. Robot receivers can use the behavior of human receivers to learn where to grasp objects handed to them by a giver, and what trajectory to navigate in space to remain safe. *Lack of background/viewpoint variation does not pose a concern*, as 3D viewpoints can be synthetically generated from HOH point clouds to emulate the 3D data from RGB-D sensors. Intended use cases in Section 1 have been expanded (highlighted) to include the above discussion.

**Benchmark evaluations (oQbw and e1Fp):**
Unfortunately, *there exist no off-the-shelf machine learning algorithms* that use human-human handover data for handover parameter estimation for HRI using 3D data such as from RGB-D sensors (or even 2D image data which may have been adapted for 3D). Only one paper, H2O by Ye et al., 2021, contains an image-based prediction of a handover parameter. Their work is limited to predicting receiver grasp. Neither their dataset nor their algorithm are public. This prevents benchmark evaluations.

Instead, we will provide a novel set of learning-driven algorithms in the revised version of the paper to serve as experimental evaluation (concern 2 of e1Fp) and provide a starting point of experimental results for future research as recommended by oQbw. They also show the benefit of the dataset for HRI. Algorithms:-
1. Predict human giver hand grasp from object point cloud (O2GGrasp): Helps robotic giver grasp algorithms be biased to grasp near the human giver’s hand. Enables robotic receiver grasp algorithms to plan their grasp to be biased in regions away from the human giver’s hand.
2. Predict giver hand motion trajectory from object point cloud (O2GTraj): Helps bias the motion of a robotic giver’s end effector to follow the path of typical human givers rather than plan unsafe trajectories, an issue of ongoing research concern in areas such as reinforcement learning for trajectory planning. Helps robotic receivers plan proactive motion.
3. Predict object transfer pose from object point cloud (O2Pose): Helps a robotic giver to orient objects in a human-receiver-preferred manner. Helps a robotic receiver plan to meet the object smoothly for seamless transfer.
4. Predict receiver hand grasp from point cloud of giver holding object (G2RGrasp): 2nd method to inform a robot receiver of where to bias grasping during object receipt.
5. Predict receiver hand motion trajectory from point cloud of giver holding object (G2RTraj): Helps inform robotic receivers to explore near human trajectories during motion planning, preventing unsafe configurations that could e.g. injure human givers.

Our work adapts point cloud generation neural networks to synthesize giver and receiver output hand point clouds for O2GGrasp/G2RGrasp, and point cloud encoding networks for pose (O2Pose) and PCA coefficients of trajectories in O2GTraj/G2RTraj. We will get results for complete point clouds and partial point clouds for multiple synthetically generated viewpoints emulating data from RGB-D sensors. Our train/test split is 3 objects per form/function bin (51 total) common to train/test, 2 per bin (34) in train only and 3 per bin (51) in test only. **We have generated results for O2GGrasp and G2RGrasp using complete point clouds with PoinTr.** Metrics are summarized in the paper Sec 5. Sample GT and predicted grasps are shown in Fig. 7. We have resources to conduct proposed experiments by Aug 29. We will reupload the paper with metrics and results. **We will also show results of robot grasp planned being guided by neural network outputs** to show future research opportunities (oQbw).

**Correctness of SAM and OpenPose (e1Fp):**
We have manually validated and corrected all SAM masks. We have manually examined all OpenPose video outputs. 95.4% of the OpenPose skeletons are well-aligned, and for only 4.6% the limb is either missing or grossly misaligned. All OpenPose outputs have confidence values available.

---

> ### Author Response · Authors · 2023-08-22
> **3D pose alignment**
>
> In response to the concern 4 of e1Fp: At this time, **we have conducted 6DOF 3D pose alignment of the object’s 3D model to all frames from G to R for all handover interactions in the dataset**, except one interaction where the hand obscures a significant portion of the object throughout. First, we conduct a frame-to-frame iterative closest point (ICP) alignment of the object point cloud to estimate inter-frame cloud-to-cloud transformations of the object point cloud for all frames, linking all frames from G to R. Next, we conduct an automated alignment of the 3D model to the object point cloud in the G frame. We sample a diverse range of orientations to ensure exhaustive SO(3) coverage. We transform each 3D model using each orientation and fine-tune the transformed model's alignment to the object point cloud using ICP. We choose the 3D model alignment with the smallest ICP distance. We use the inter-frame transformations to align the 3D model to all frames from G to R. Example frames from the alignment are shown in the paper in Fig. 4. We also show example videos of the alignment at the shared webpage. We have now updated various regions of the paper, including the list of contributions in Sec 1 to discuss the 6DOF pose. Due to our small team size, we do not have the capacity to align hand poses at this time, however, the object poses enable us to create neural networks for object pose estimation at transfer (O2Pose).

---

> > ### Author Response · Authors · 2023-08-26
> > **Updated paper with benchmarks for neural networks for giver grasp, receiver grasp, and transfer orientation prediction**
> >
> > As an update to all reviewers, especially e1Fp and oQbw, we have now uploaded a copy of the paper containing **benchmark results** from neural networks trained to perform
> > 1. giver grasp prediction from an object point cloud (o2gg),
> > 2. receiver grasp prediction from object and giver point cloud (g2rg),
> > 3. and transfer orientation prediction from object point clouds (o2or).
> >
> > o2gg and g2rg have been done for complete and partial point clouds, and o2or for complete point clouds.  o2or is currently training for partial data. As discussed in the paper, we are retargeting trajectory prediction to perform proactive forecasting of the receiver trajectory from knowledge of the giver trajectory (g2rt), which may be acquired by existing object tracking techniques, to demonstrate the value of our dataset in modeling timing and coordination during handover. We expect to complete g2rt by early next week, and we are aiming to have all experiments by 24 hours prior to the author/reviewer discussion deadline.
> >
> > Ground truth (GT) metrics are computed for all completed networks and reported in Table 3.  Since plausible handover parameters need not correspond to GT (e.g., a person can grasp an object plausibly at a location not in the GT), we also report the best GT metric to all occurrences of the object in the dataset (the best GT metric is not applicable to the percentage overlap). Best GT has been completed for the g2rg network and is currently running for the remaining networks. Empty cells in Table 3 will be filled once networks have completed running and summary metrics are obtained. Since the prior submission a few days ago, figures have been rearranged, and additional qualitative results for o2gg, g2rg, and o2or are shown in Figure 9. Once all material is in, we plan to compactify the dataset collection and data analysis sections by moving some material to the supplementary, in order to ensure that the paper fits within the 10+refs-page limit.

---

> > > ### Author Response · Authors · 2023-08-28
> > > **Final discussion-phase update**
> > >
> > > Thank you to all the reviewers for their insightful discussions, and we greatly appreciate the reviewers noticing the strengths of our work and observing our benchmark results. We appreciate reviewers oQbw and e1Fp updating their score. At this time, we have uploaded a copy of the paper and the supplementary with the following summary changes in response to reviewer feedback:
> > >
> > > 1. Benchmark results supporting intended use cases for o2gg (Object to giver grasp), g2rg (giver+object to receiver grasp), and o2or (object to transfer pose orientation) using complete and partial point cloud data, as well as g2rt (giver to receiver trajectory prediction) from tracks of giver and receiver. Benchmark results are now in Section 4. Metrics in Table 3 are complete, and visual results are shown as well. Additional visual results are uploaded in the new copy of the supplementary.
> > > 2. Clarification on the intended use cases in the Introduction and Section 4. Potential future work that the experimental results can lead to has been discussed in Section 5, retitled 'Discussion'.
> > > 3. Reorganization of Section 3 to have subsections summarizing dataset philosophy, dataset collection, and dataset analysis (previously a separate section) to enable inclusion of benchmark results.
> > >
> > > Additional content added in response to feedback is highlighted in the uploaded copy of the paper.

---

### Decision · Program_Chairs · 2023-09-22

**Decision:**

Accept (Poster)

**Comment:**

This submission introduces a large-scale real-captured hand-object-human handover dataset. It initially got mixed positive and negative reviews. One common major concern was a lack of demonstrated applications. The initial submission was mainly only about how to collect the dataset and show statistics of the dataset without actual demonstration of any usage of this dataset. In the rebuttal, the authors showed multiple benchmark results, and all reviewers are satisfied with the new results. I highly recommend the authors to include the newly introduced benchmark results.